# The bridge-like lipid transport protein VPS13C/PARK23 mediates ER–lysosome contacts following lysosome damage

Xinbo Wang [1,2,3,4,5], Peng Xu [1,2,3,4,5], Amanda Bentley-DeSousa[1,2,4,5], William Hancock-Cerutti [1,2,3,4,5], Shujun Cai [1,2,3,4,5], Benjamin T. Johnson [1,2,3,4,5], Francesca Tonelli [5,6], Lin Shao [7], Gabriel Talaia[1,2,4], Dario R. Alessi [5,6], Shawn M. Ferguson [1,2,4,5] & Pietro De Camilli [1,2,3,4,5] ✉

Based on genetic studies, lysosome dysfunction is thought to play a pathogenetic role in Parkinson's disease. Here we show that VPS13C, a bridge-like lipid-transport protein and a Parkinson's disease gene, is a sensor of lysosome stress or damage. Following lysosome membrane perturbation, VPS13C rapidly relocates from the cytosol to the surface of lysosomes where it tethers their membranes to the ER. This recruitment depends on Rab7 and requires a signal at the damaged lysosome surface that releases an inhibited state of VPS13C, which hinders access of its VAB domain to lysosome-bound Rab7. Although another Parkinson's disease protein, LRRK2, is also recruited to stressed or damaged lysosomes, its recruitment occurs at much later stages and by different mechanisms. Given the role of VPS13 proteins in bulk lipid transport, these findings suggest that lipid delivery to lysosomes by VPS13C is part of an early protective response to lysosome damage.

Extensive genetic studies have identified many genes whose mutations cause or increase the risk of Parkinson's disease (PD). Among them is *VPS13C* (*PARK23*)[1–3], which encodes a member of a family of bridge-like lipid-transport proteins that act at contacts between intracellular membranes[4]. These are rod-like proteins with a hydrophobic groove running along their length where lipids are thought to flow unidirectionally from one bilayer to another[5–8]. The human genome encodes four VPS13 proteins, VPS13A, VPS13B, VPS13C and VPS13D, which have distinct, partially overlapping localizations at membrane contact sites[9–14], with VPS13C being localized at contacts between the endoplasmic reticulum (ER) and late endosomes/lysosomes (hence referred to as endolysosomes) as well as at ER–lipid droplet contacts[10,15].

Human VPS13C is a very large protein whose approximately 30-nm-long rod-like core is flanked at its carboxy-terminal side by folded modules, namely a Vps13-adaptor-binding (VAB) domain, a WWE domain, an ATG2C domain (a bundle of four amphipathic helices so-called due to a similarity to the C-terminal region of ATG2) and a C-terminal PH domain (Fig. 1a)[16,17]. VPS13C binds the ER via an interaction of its amino-terminal portion with the ER membrane protein VAP (an FFAT motif-dependent interaction) and endolysosomes via an interaction of its VAB domain with Rab7 (ref. 10). It can also bind lipid droplets via its ATG2C domain. The bridge-like arrangement of VPS13C in situ at contacts between the ER and lysosomes has been supported by cryo-electron tomography (cryo-ET)[17]. Building on insights from other bridge-like lipid-transport proteins, also referred to as

[1]Department of Neuroscience, Yale University School of Medicine, New Haven, CT, USA. [2]Department of Cell Biology, Yale University School of Medicine, New Haven, CT, USA. [3]Howard Hughes Medical Institute, Yale University School of Medicine, New Haven, CT, USA. [4]Program in Cellular Neuroscience, Neurodegeneration, and Repair, Yale University School of Medicine, New Haven, CT, USA. [5]Aligning Science Across Parkinson's (ASAP) Collaborative Research Network, Chevy Chase, MD, USA. [6]MRC Protein Phosphorylation and Ubiquitylation Unit, School of Life Sciences, University of Dundee, Dundee, UK. [7]Center for Neurodevelopment and Plasticity, Wu Tsai Institute, Yale University, New Haven, CT, USA. ✉e-mail: pietro.decamilli@yale.edu

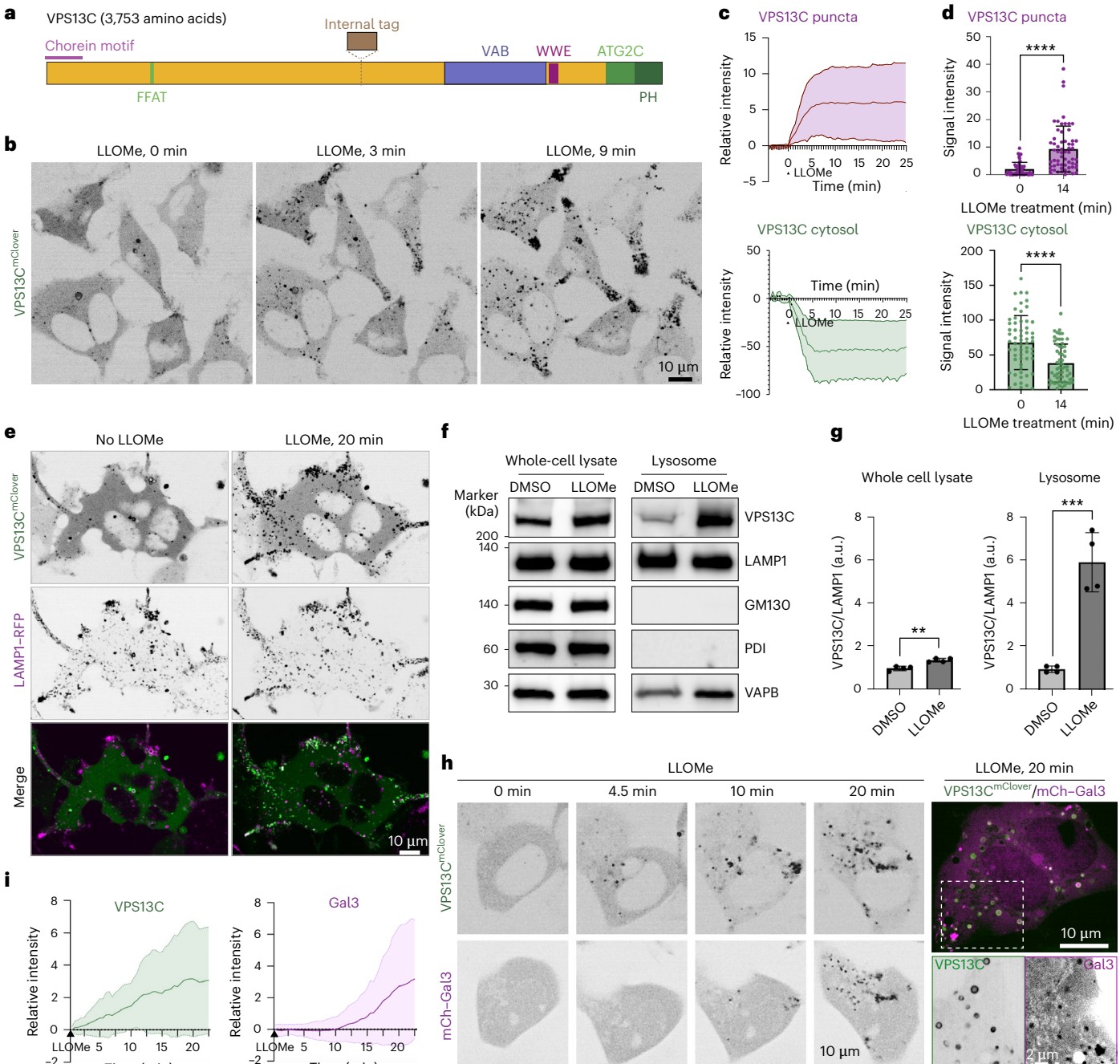

**Fig. 1 | VPS13C is acutely recruited to damaged lysosomes. a**, Domain organization of human VPS13C. **b**,**c**, Fluorescence image time series of live VPS13C^mClover-Flp-In cells showing rapid VPS13C recruitment to damaged lysosomes following treatment with 1 mM LLOMe. **c**, Intensity of the punctate (top) and cytosolic (bottom) VPS13C^mClover fluorescence per cell before and after treatment with 1 mM LLOMe is shown for a representative experiment; *n* = 21 cells were analysed. **d**, Intensity of punctate (top) and cytosolic (bottom) VPS13C^mClover fluorescence per cell before and after treatment with 1 mM LLOMe for 14 min; *n* = 60 cells collected from three biological replicates. **e**, Fluorescence images of live VPS13C^mClover-Flp-In cells co-expressing the lysosome marker Lamp1–RFP before and after treatment with 1 mM LLOMe. **b**,**e**, The experiments were repeated three times with similar results. **f**, Western blot analysis for the indicated proteins in whole cell lysates (left) and SPION-purified lysosomal fractions (right) of cells treated with 1 mM LLOMe or dimethylsulfoxide (DMSO) for 20 min.

**g**, Quantification of the data in **f**. Bars show the normalized value relative to DMSO; *n* = 4 biological replicates; \*\**P* = 0.0013, \*\*\**P* = 0.0004; a.u., arbitrary units. **d**,**g**, Error bars represent the s.d. **h**, Fluorescence image time series of live VPS13C^mClover-Flp-In cells co-expressing mCh–Gal3 following treatment with 1 mM LLOMe. A merge field of the mClover and mCh fluorescence at 20 min is shown (right). Individual channel images of the dashed box are shown at the bottom. **i**, Intensity of the punctate fluorescence of VPS13C^mClover and mCh–Gal3 in **h** before and after 1 mM LLOMe treatment; *n* = 58 VPS13C and 52 Gal3 cells collected from three biological replicates. **c**,**i**, Data are the normalized fluorescence relative to time 0 (LLOMe treatment start); mean ± s.d. represented by the solid line and shaded area, respectively. **d**,**g**, Data were compared using a two-sided Student's *t*-test; \*\**P* < 0.01, \*\*\**P* < 0.001, \*\*\*\**P* < 0.0001. All the individual channel images in this figure are shown as inverted greys. Numerical source data and unprocessed blots are provided.

RGB proteins because of the repeating β-groove elements that constitute their rod-like core[7,8], VPS13C is hypothesized to mediate net lipid (primarily phospholipid) transport from the ER to endolysosomes[11]. However, its precise physiological function remains elusive.

Loss-of-function mutations in *VPS13C* cause PD[1]. Genetic deletion of *VPS13C* in HeLa cells or induced pluripotent stem cell-derived human neurons results in altered lysosome homeostasis[15,18]. Notably, in *VPS13C*-knockout (KO) HeLa cells the cGAS–STING innate immunity pathway[19] is activated, probably reflecting impaired STING degradation in dysfunctional lysosomes and escape of mitochondrial DNA into the cytosol[15]. These findings indicate a potential role for VPS13C in maintaining the integrity of the lysosomal membrane and possibly also in membrane repair. They are especially relevant in the context of PD, as several PD-related genes encode proteins implicated in lysosomal function[20-22]. These include the gene encoding LRRK2, the most frequently mutated protein in familial PD[23,24]. LRRK2 can be acutely recruited to lysosomes in response to various manipulations that induce perturbation of their membranes, including treating cells with L-leucyl-L-leucine methyl ester (LLOMe), a lysosomotropic agent that accumulates in lysosomes and is converted to membranolytic metabolites in the lysosome lumen[25,26]. Several factors other than LRRK2 are recruited to endolysosomes in response to damage. These include ESCRT proteins that function as key players in membrane-repair mechanisms[27-31]. They also include the ER-anchored OSBP/ORP family lipid transport proteins that are recruited in response to the generation of phosphatidylinositol-4-phosphate (PI4P) on endolysosome membranes in the phosphoinositide-initiated membrane tethering and lipid transport (PITT) pathway[32-34].

VPS13C is normally present only in a subset of endolysosomes, suggesting the requirement of specific conditions for such localization. We found that the formation of VPS13C-dependent ER–endolysosome tethers is part of an early response to lysosome damage and is differentially controlled relative to the recruitment of ESCRT, OSBP/ORP family proteins and LRRK2. Absence of VPS13C makes lysosomes more sensitive to LLOMe-induced perturbation. Our results also show that localization of VPS13C at the interface between the ER and lysosomes requires binding of its ATG2C domain to the damaged lysosome membrane, leading to the release of an auto-inhibition that prevents access of VPS13C to Rab7 on the surface of endolysosomes.

## Results

### VPS13C is acutely recruited to damaged lysosomes

To track the localization of VPS13C in cells, we generated a Flp-In TREx 293 stable cell line expressing VPS13C tagged with mClover at an internal site under the control of tetracycline (referred to henceforth as VPS13C[mClover-Flp-In] cells; Fig. 1a and Extended Data Fig. 1a). Western blot analysis of the homogenates of these cells with anti-VPS13C revealed that, under the conditions used for our experiments (24 or 48 h in the presence of 0.1 µg ml[-1] tetracycline), no global overexpression of VPS13C occurred as the levels of endogenous VPS13C were reduced in parallel with expression of exogenous VPS13C[mClover] (Extended Data Fig. 1b).

Confocal imaging of these VPS13C[mClover]-expressing cells showed puncta or doughnut-like structures (Fig. 1b), which corresponded to a small subset of lysosomes, as confirmed by co-localization with the lysosome marker LAMP1–RFP (Fig. 1e). An additional pool of VPS13C was diffusely distributed throughout the cytosol. However, following the addition of 1 mM LLOMe, a lysosomotropic agent that causes lysosomal membrane perturbation[35], a rapid (within minutes) and massive recruitment of VPS13C[mClover] to the majority of LAMP1–RFP[+] organelles was observed (Fig. 1b–e). LAMP1[+] lysosomes were concentrated at the tips of cell processes in this cell line and their localization was unaffected by exposure to LLOMe (Fig. 1e). Recruitment of VPS13C to lysosomes correlated with lysosomal membrane damage, as pre-incubation of cells with the cathepsin C inhibitor E64d, the enzyme that processes LLOMe into membranolytic polymers, inhibited VPS13C recruitment (Extended Data Fig. 1c).

To biochemically validate the recruitment of VPS13C to lysosomes in response to LLOMe as well as determine whether such recruitment is also observed for endogenous untagged VPS13C, we used a previously described technique for the purification of lysosomes. The method is based on incubation of live cells with dextran-conjugated superparamagnetic iron oxide nanoparticles (SPIONs) to allow endocytic uptake, followed by a chase to allow traffic of the particles to lysosomes, and subsequent cell lysis and recovery of lysosomes on a magnetic column[15,36,37]. Western blot analysis of the isolated material revealed an abundant presence of LAMP1, but not GM130 (Golgi marker) or PDI (a luminal ER marker), confirming the selective isolation of lysosomes (Fig. 1f). Cells treated with LLOMe had a very robust enrichment of VPS13C relative to LAMP1 (Fig. 1f,g), confirming the imaging experiments. An additional slight increase in the ER protein VAP is in line with the recruitment of VPS13C (Fig. 1f), as VPS13C binds VAP and fragments of ER membranes are expected to co-purify with endolysosomes (Fig. 2b and subsequent results).

### VPS13C recruitment precedes Gal3 recruitment

Expression of fluorescently tagged galectin 3 (Gal3) has been used as a tool to detect severely damaged lysosomes[27,38]. Galectin 3 is a β-galactoside-binding lectin that remains soluble in the cytosol of healthy cells. However, it rapidly translocates to damaged lysosomes, where it binds the glycoprotein-rich luminal surface of their membranes to help coordinate lysosomal repair and removal processes[38-40]. When transiently expressed in VPS13C[mClover-Flp-In] cells, mCherry (mCh)-tagged Gal3 (mCh–Gal3) localized at the cytosol under basal conditions. Treatment with LLOMe induced the formation of Gal3 puncta, indicating its recruitment to damaged lysosomes (Fig. 1h). Recruitment of VPS13C[mClover] preceded the recruitment of Gal3 to lysosomes by several minutes (Fig. 1h,i), which implies that VPS13C senses a change in the properties of the lysosomal membrane that precedes severe membrane rupture.

An early response to lysosome damage by LLOMe, which also precedes Gal3 recruitment, is the recruitment of the ESCRT complex[27,28]. A comparison of the LLOMe-induced recruitment of VPS13C (VPS13C[mClover]) and the ESCRT-III component IST1 (IST1-Apple) suggested a roughly similar time course of the two proteins (Extended Data Fig. 2a). However, whereas the LLOMe-triggered IST1 recruitment was inhibited by the chelation of cytosolic Ca[2+] by BAPTA, consistent with its dependence on the leakage of Ca[2+] from damaged lysosomes[27,41,42], VPS13C recruitment was not blocked by BAPTA (Extended Data Fig. 2b). Thus, although the recruitment of both VPS13C and ESCRTIII represent a very early response to lysosome damage, their recruitment seems to be mediated by different mechanisms.

### VPS13C recruited to lysosomes is bound to the ER

The proposed bridge-like lipid transport function of VPS13C at ER–endolysosome contacts implies its simultaneous binding to both the ER and endolysosomes. Binding to the ER relies on the interaction of its FFAT motif with the transmembrane ER protein VAP (VAPA and VAPB), whereas binding to the endolysosomal membrane requires the interaction of its VAB domain with Rab7 (refs. 10,15; Fig. 2a). If LLOMe-induced recruitment of VPS13C to lysosomes is aimed at allowing a flux of lipids from the ER to the damaged membranes of these organelles, the recruitment of VPS13C to lysosomes should be accompanied by the recruitment of the ER via VAP. Consistent with this hypothesis, following LLOMe treatment of VPS13C[mClover-Flp-In] cells with additional expression of mCh-tagged VAPB (mCh–VAPB), the VPS13C recruitment was accompanied by VAP enrichment at endolysosomes (Fig. 2b).

### Rab7 dependence of VPS13C recruitment

Binding of VPS13C to lysosomes was shown to be dependent on the interaction of its VAB domain with Rab7 and to be abolished by expression of dominant negative Rab7 (ref. 15). Accordingly, the accumulation of VPS13C at lysosomes in response to LLOMe was significantly

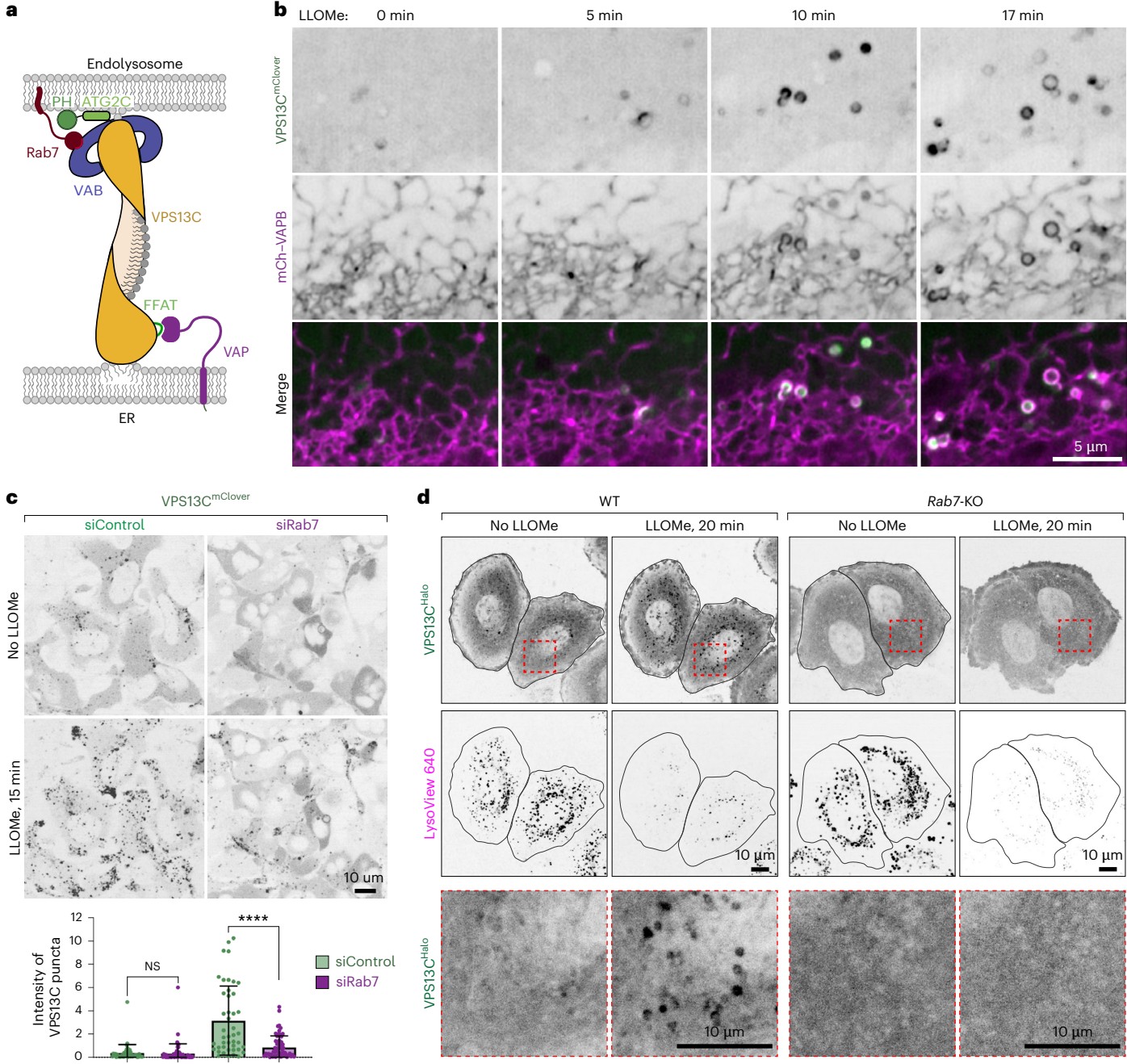

**Fig. 2 | VPS13C functions at the ER–endolysosome membrane contact sites.**
**a**, Cartoon depicting VPS13C localized at the ER–endolysosome membrane contact sites. **b**, High-magnification time-series images of VPS13C^mClover-Flp-In cells co-expressing exogenous mCh–VAPB following treatment with 1 mM LLOMe. **c**, Fluorescence images of live VPS13C^mClover-Flp-In cells showing VPS13C^mClover localization in control (siControl, control siRNA) and *Rab7*-knockdown (siRab7, siRNA to *Rab7*) cells before and after treatment with 1 mM LLOMe (top). Fluorescence intensity of VPS13C^mClover puncta signals per cell before and after LLOMe treatment (bottom). Error bars represent the s.d.; *n* = 44 siControl and 60 siRab7 cells collected from three biological replicates. Data were compared using a two-sided Student's *t*-test; NS, not significant (*P* = 0.7437) and ****P < 0.0001. **d**, Fluorescence images of live WT and *Rab7*-KO HeLa cells, expressing VPS13C^Halo and labelled with LysoView 640 (a luminal marker of acidic lysosomes), before and after treatment with 1 mM LLOMe. Black lines represent cell outlines. Magnified views of the boxed regions in the VPS13C^Halo channel are shown (bottom). **b**,**d**, The experiments were repeated three times with similar results. All the individual channel images in this figure are shown as inverted greys. Numerical source data are provided.

reduced in VPS13C^mClover-Flp-In cells following *Rab7* knockdown (Extended Data Fig. 3a) compared with wild-type (WT) cells (Fig. 2c). Moreover, puncta of VPS13C fluorescence reflecting accumulation of VPS13C at lysosomes no longer occurred either under basal conditions or after LLOMe treatment of *Rab7*-KO HeLa cells[43] (Fig. 2d) expressing VPS13C^Halo and pre-incubated with LysoView 640. Occurrence of lysosome damage in these cells after LLOMe treatment was confirmed by the loss of LysoView 640 signal. Interestingly, VPS13C protein levels were significantly elevated in *Rab7*-KO cells (Extended Data Fig. 3b,c), possibly as a compensatory response. We conclude that both basal binding of VPS13C to lysosomes and its enhanced recruitment following lysosome damage are Rab7-dependent events. These findings prompted us to examine how Rab7-dependent recruitment of VPS13C is regulated.

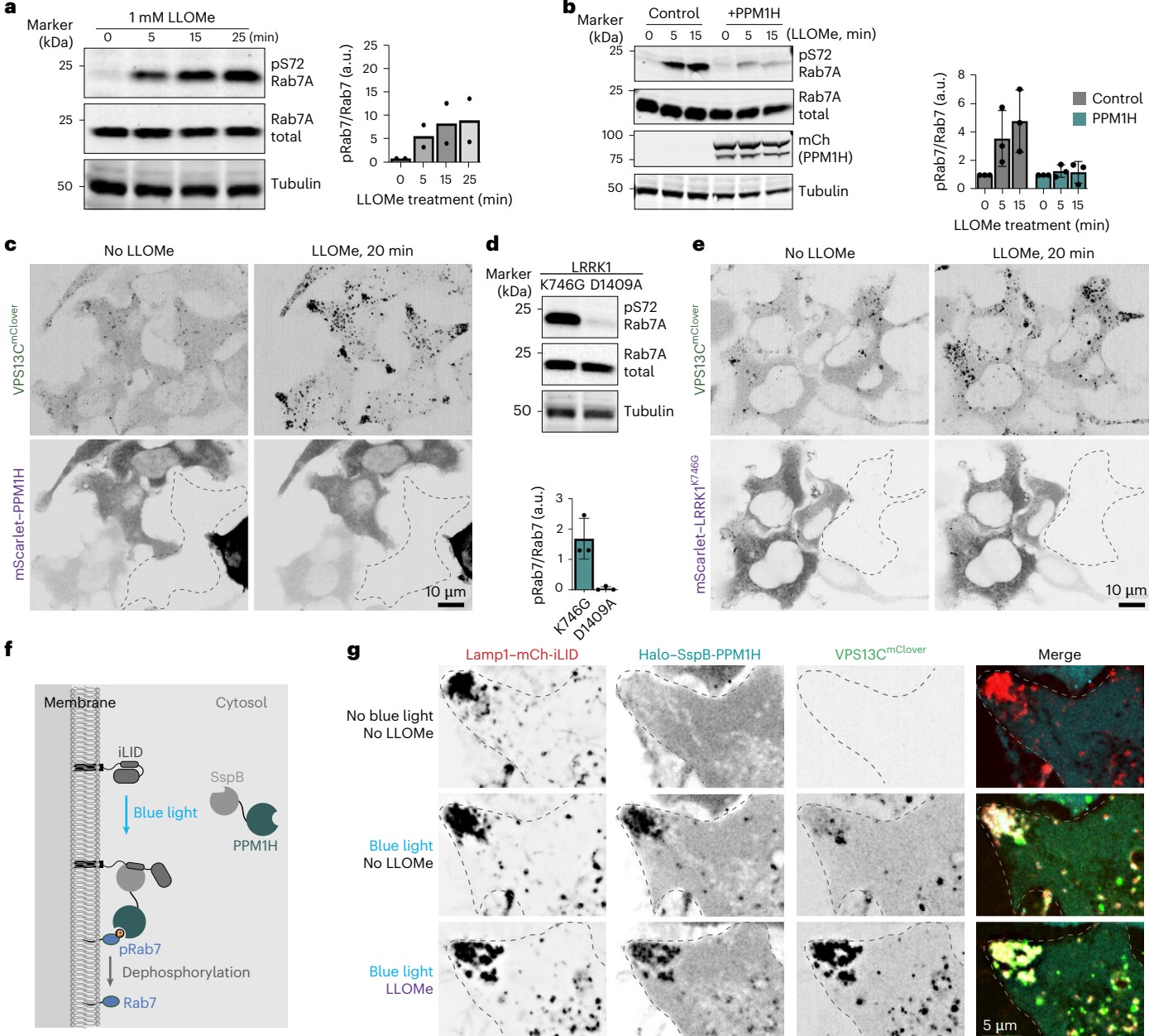

**Fig. 3 | Rapid Rab7 phosphorylation induced by LLOMe is unnecessary for VPS13C recruitment. a,b**, Western blot analysis of the indicated proteins in whole-cell lysates of VPS13C^mClover-Flp-In cells at different time points after treatment with 1 mM LLOMe (left). Protein quantification (right). The bars show normalized values relative to time 0; *n* = 2 (**a**) and 3 (**b**) biological replicates. **b**, Subsets of cells were transfected with mScarlet–PPM1H. **c**, Fluorescence images of live VPS13C^mClover-Flp-In cells with mScarlet–PPM1H co-expression before and after treatment with 1 mM LLOMe. Cells not expressing mScarlet–PPM1H are outlined by dashed lines. **d**, Western blots for total Rab7 and Ser72-phosphorylated Rab7 (tubulin was used as a loading control) of whole-cell lysates of VPS13C^mClover-Flp-In cells co-expressing constitutively kinase-active (mScarlet–LRRK1^K746G) or kinase-dead (mScarlet-LRRK1^D1409A) LRRK1 mutants (top). Protein quantification (bottom). The bars show normalized values relative to the signal

obtained with kinase-dead LRRK1; *n* = 3 biological replicates. **a,b,d**, The error bars represent the s.d. **e**, Fluorescence images of live VPS13C^mClover-Flp-In cells with co-expression of the constitutively kinase-active LRRK1 mutant (mScarlet-LRRK1^K746G) before and after treatment with 1 mM LLOMe. Cells not expressing mScarlet–LRRK1^K746G are outlined by dashed lines. **f**, Schematic of the iLID light-dependent protein heterodimeric system used in **g. g**, Fluorescence images of live VPS13C^mClover-Flp-In cells with co-expression of exogenous Lamp1–mCh-iLID (bait) and Halo–SspB-PPM1H (prey) before and after blue light illumination and treatment with 1 mM LLOMe. Dashed lines represent cell outlines. **c,e,g**, The experiments were repeated three times with similar results. pS72, phosphorylation of Rab7 at Ser72. All the individual channel images in this figure are shown as inverted greys. Numerical source data and unprocessed blots are provided.

## VPS13C recruitment is independent of Rab7 phosphorylation

Phosphorylation of Rab GTPases has emerged as an important factor in the regulation of their functions[44–46]. Specifically, phosphorylation of Rab7 at Ser72 functions as a phosphoswitch that determines its interactions with distinct downstream effectors[47,48]. TBK1 and LRRK1 mediate

this phosphorylation event, whereas its dephosphorylation is regulated by the phosphatase PPM1H[47,49,50]. We found that, Rab7 undergoes fast Ser72 phosphorylation in VPS13C^mClover-Flp-In cells following LLOMe treatment (Fig. 3a), with the same time course of VPS13C recruitment shown in Fig. 1. To investigate potential contributions of LRRK1 and TBK1 to

this LLOMe-dependent increase in Rab7 phosphorylation at Ser72, we assessed the impact of these kinases in KO mouse embryonic fibroblasts (MEFs)[49] and found that such increases seemed to be reduced by *LRRK1* KO but not by the double KO of *TBK1* and its paralogue *IKKe* (Extended Data Fig. 4). On the other hand, in HeLa cells, TBK1, which can also be activated by LLOMe[51], was found to be responsible for approximately 50% of the increase in Rab7 Ser72 phosphorylation induced by LLOMe[43], suggesting the occurrence of cell-specific differences in this event.

To test the role of Rab7 Ser72 phosphorylation in the regulation of VPS13C recruitment, we implemented multiple strategies. First, we transiently overexpressed PPM1H[49] into VPS13C[mClover-Flp-In] cells to retain Rab7 in its dephosphorylated state. Western blot analysis confirmed a strong reduction in LLOMe treatment-induced Rab7 phosphorylation (Fig. 3b), but recruitment of VPS13C occurred both in cells expressing PPM1H and in their neighbouring PPM1H⁻ cells (Fig. 3c). Conversely, overexpression of a constitutively kinase-active LRRK1 mutant, LRRK1[K746G] (but not the kinase-dead LRRK1[D1409A])[49], robustly induced Rab7 Ser72 phosphorylation under basal conditions (Fig. 3d) but failed to increase VPS13C recruitment both under basal conditions and following LLOMe treatment (Fig. 3e). To exclude the possibility that these negative results may reflect adaptive changes following expression of PPM1H or mutant LRRK1, we acutely manipulated Rab7 phosphorylation at the lysosomal surface. To this aim, we employed the improved[52] light-induced dimerization (stringent starvation protein B, SspB; iLID) system[53] to recruit PPM1H to lysosome membranes in response to blue light (Fig. 3f). We observed rapid translocation of cytosolic SspB-linked PPM1H to lysosomes following blue light illumination (Fig. 3g(middle)) but this translocation did not prevent the prompt recruitment of VPS13C following LLOMe treatment (Fig. 3g (bottom)). Together, although these results demonstrate a robust increase in Rab7 phosphorylation following LLOMe treatment, such a phosphorylation reaction is not required for VPS13C recruitment.

### Full-length VPS13C access to Rab7 is regulated

As mentioned earlier, binding of VPS13C to lysosomes depends on the interaction of its VAB domain with Rab7 (refs. 10,15). Although VPS13C recruitment to lysosomes by LLOMe could be explained by an enhanced activated state (GTP-loaded) of Rab7, a discrepancy that we have observed between the localization of the VAB domain fragment of VPS13C and the localization of full-length VPS13C contests this possibility. Before LLOMe addition, VPS13C was enriched only on a small subset of lysosomes in VPS13C[mClover-Flp-In] cells (Fig. 1e). In contrast, when the mCh-tagged VAB domain (Fig. 4a), which mediates the interaction with Rab7, was expressed in these cells, robust co-localization of this construct with the great majority of organelles positive for the lysosomal marker LysoView 640 occurred even under basal conditions (Fig. 4b(left)). This massive localization of the VAB domain on lysosomes even under basal conditions was further validated in HeLa cells (Extended Data Fig. 5a). Importantly, this association of the VAB domain with lysosomes was lost in *Rab7*-KO HeLa cells (Extended Data Fig. 5a).

Minutes after LLOMe treatment, when loss of LysoView 640 signal from most lysosomes signalled their damage, much of the VAB domain was shed and this coincided with the accumulation on lysosomes of full-length VPS13C[mClover] with a reduction of its cytosolic pool (Fig. 4b(left)). Interestingly, the few lysosomes that remained strongly positive for the LysoView 640 signal even after LLOMe treatment also retained a strong VAB signal and failed to recruit full-length VPS13C[mClover] (Fig. 4c). These results indicate that access of full-length VPS13C, but not of its VAB domain alone, to Rab7 is regulated.

To further explore this regulation, we monitored the dynamics of co-expressed full-length VPS13C and the VAB-domain-only construct in hTERT RPE-1 (RPE1) cells, that is, cells where the presence of large lysosomes facilitates their tracking over time. Even in these cells, strong lysosomal binding of the VAB domain, but only scattered

lysosomal binding of full-length VPS13C, was observed under basal conditions (Fig. 4d), whereas a swap occurred after LLOMe treatment, with an increased lysosome association of the full-length protein and decreased association of the VAB-domain-only (Fig. 4d(right)). As binding of Rab effectors to Rab reflects a dynamic equilibrium of binding/ unbinding reactions, one potential explanation for this finding is that the recruitment of the ER (Fig. 2b) to the lysosomal membranes creates a barrier that prevents rebinding of the VPS13C VAB domain once the ER has covered the endolysosomal membrane. It was recently shown that not only endogenous VPS13C but also other ER–lysosomal tethers are recruited to lysosomes in response to LLOMe[32,33] and can thus contribute to this ER recruitment-dependent effect. In fact, a decrease in VAB binding following LLOMe treatment did not require overexpression of VPS13C (Extended Data Fig. 5b(left)). Collectively, these findings suggest that the availability of active Rab7 is not limiting and that in full-length VPS13C access of the VAB domain to Rab7 binding may be subject to a LLOMe-regulated mechanism.

### An intramolecular regulation controls VPS13C access to Rab7

One factor that could limit access of full-length VPS13C to lysosome-associated Rab7 in the absence of lysosome damage is an intramolecular interaction blocking the Rab7 binding site in VPS13C (Fig. 4a). The VAB domain of VPS13C is flanked by other folded modules (Fig. 2a), a WWE domain, the ATG2C domain and a PH domain, which project out of the rod-like core of VPS13C[17]. The position of these modules relative to each other and to the rod-like core is not fixed, as suggested by low-resolution negative-staining electron microscopy analysis of yeast VPS13 (ref. 54). To test for a potential inhibitory interaction between the VAB domain and other folded modules that flank the rod-like core of VPS13C at its C-terminal region, we expressed the mCh-tagged C-terminal fragment of VPS13C (mCh-VPS13C[C-ter]), which comprises the VAB, WWE, ATG2C and PH domains, in addition to a small fragment of the rod-like core in VPS13C[mClover-Flp-In] cells (Fig. 4a,b(right)). The VPS13C[C-ter] fragment acted similarly to full-length VPS13C, with few lysosomes positive for this construct and a predominant cytosolic localization under basal conditions (Fig. 4b(right) and Extended Data Fig. 5c(left)). These results are consistent with the idea that some structural element within the C-terminal fragment of VPS13C obstructs access of the VAB domain to Rab7. However, in contrast to what was observed with full-length VPS13C, even this low binding to lysosomes observed under basal conditions was mostly lost following exposure to LLOMe, when robust recruitment of full-length VPS13C was observed (Fig. 4b(right)). Similar results were observed when VPS13C[Halo] and the mCh–VPS13C[C-ter] construct were co-expressed in RPE1 cells (Fig. 4e and Extended Data Fig. 5b,c(right)). These results support the hypothesis that recruitment of the ER to lysosomes reduces the association of cytosolic factors bound to Rab7 with their surface by hindering their access to such surface.

Based on these findings we suggest that an intramolecular interaction within the C-terminal fragment of VPS13C prevents binding of the VAB domain to Rab7 and that a signal(s) generated by lysosomal damage is required to release such inhibition.

### The ATG2C domain of VPS13C detects lysosome-membrane damage

To directly test whether the C-terminal fragment of VPS13C, which contains the ATG2C and PH domains (Fig. 1a), blocks VAB domain access to Rab7, we generated a VPS13C construct lacking these domains, VPS13C-Δ(ATG2C-PH) (Fig. 5a). Expression of this construct with an internal Halo tag, VPS13C-Δ(ATG2C-PH)[Halo], in RPE1 cells revealed strong lysosomal binding (Fig. 5b) under basal conditions, comparable to the binding of VAB-domain-only (Fig. 4d). This was in contrast to the faint lysosomal binding observed for full-length VPS13C[Halo] under these conditions (Figs. 4d,e and 5b). We conclude that the C-terminal fragment of VPS13C, which comprises the ATG2C and PH domains, acts

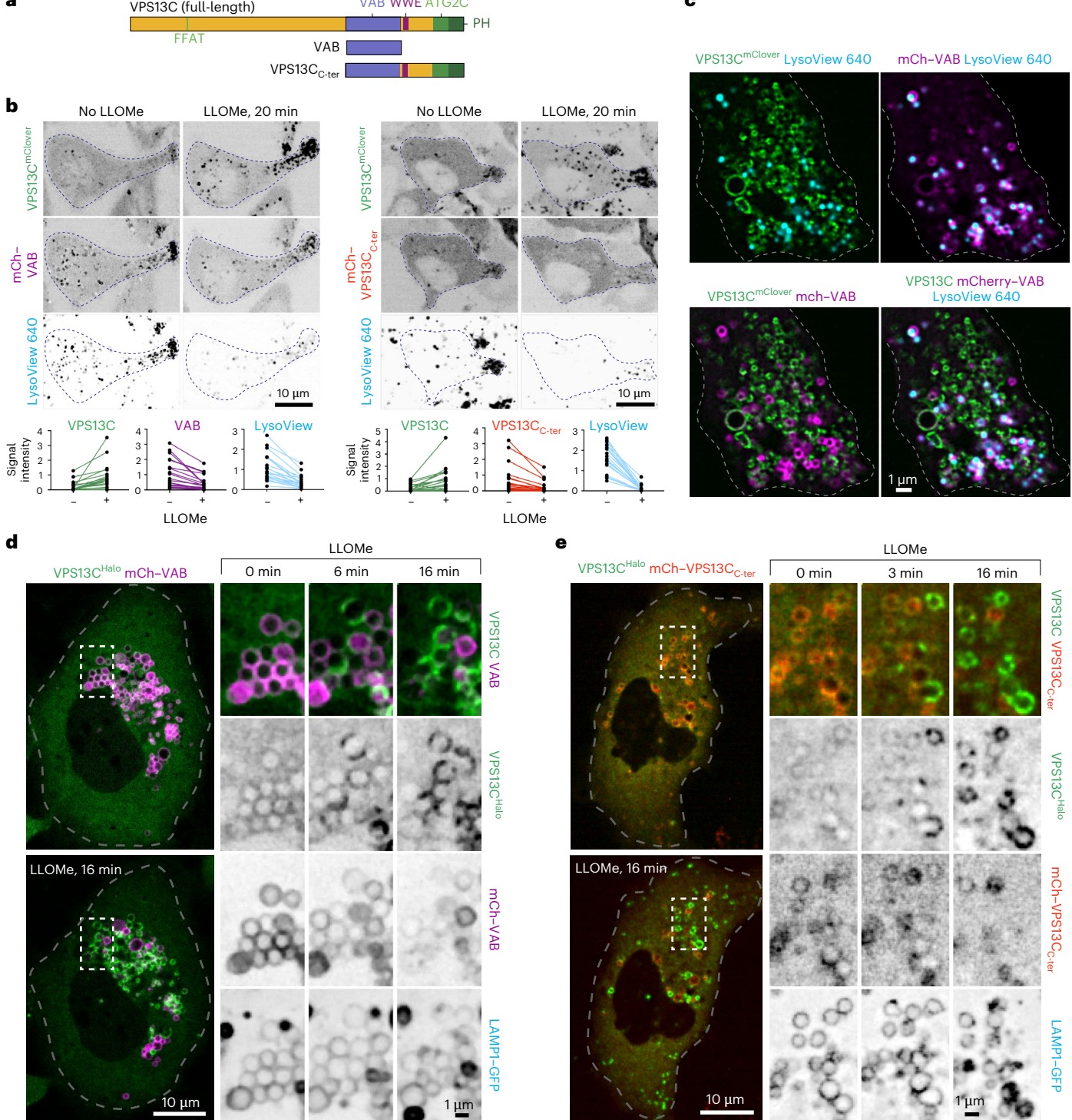

**Fig. 4 | An intramolecular regulation controls access of full-length VPS13C to Rab7. a**, Domain organization of full-length VPS13C. Deletion constructs used for the experiments of this figure are also indicated. **b**, Fluorescence images of live VPS13C^mClover-Flp-In cells with co-expression of the mCh–VAB domain (top left) or mCh–VPS13C_C-ter (top right), and labelled with LysoView 640, before and after treatment with 1 mM LLOMe. Levels of the punctate fluorescence from individual channels in a representative experiment (bottom). Each line represents the average intensity of the indicated signals from the same cell before and after LLOMe treatment; *n* = 20 cells were analysed. **c**, High-magnification fluorescence images of live VPS13C^mClover-Flp-In cells with co-expression of mCh–VAB 20 min after the addition of 1 mM LLOMe. Almost no overlap between mCh–VAB and VPS13C^mClover fluorescence was observed. **d,e**, Fluorescence images of live RPE1 cells co-expressing exogenous VPS13C^Halo and either mCh–VAB (**d**) or mCh–VPS13C_C-ter (**e**) before and after treatment with 1 mM LLOMe. Time series of individual channels from the boxed regions are shown (right). **b–e**, Blue (**b**) and grey (**c–e**) dashed lines in the images represent cell outlines. The experiments were repeated three times with similar results. Numerical source data are provided.

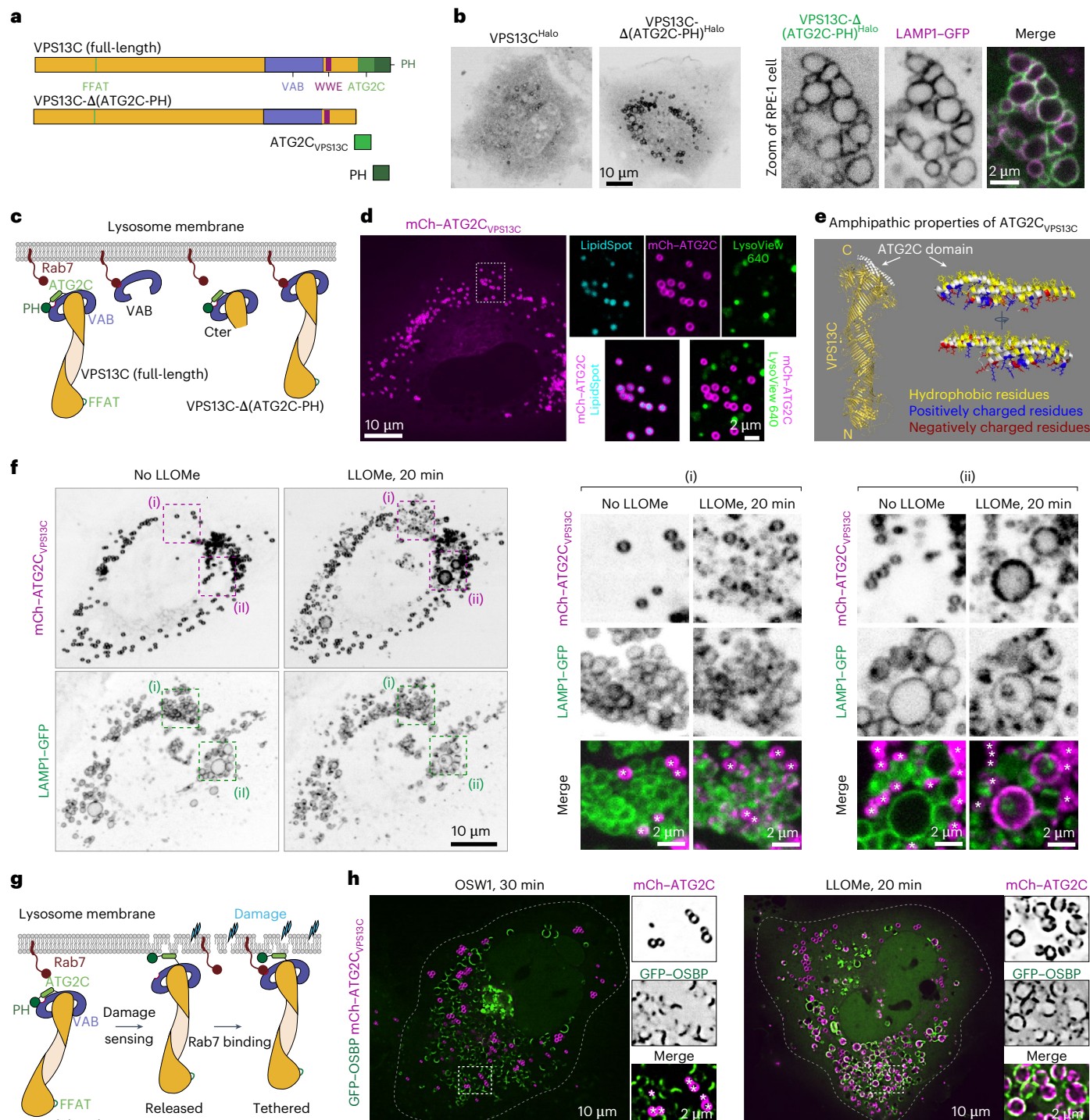

**Fig. 5 | The ATG2C domain of VPS13C detects lysosome-membrane damage.**
**a**, Domain organization of full-length VPS13C (top) as well as the domains of VPS13C used for the experiments of this figure (bottom). **b**, Fluorescence images of live RPE1 cells expressing full-length VPS13C^Halo or VPS13C-Δ(ATG2C-PH)^Halo under basal conditions (left). Magnified view of a region of a RPE1 cells showing co-localization of VPS13C-Δ(ATG2C-PH)^Halo with Lamp1–GFP (right). **c**, Cartoon depicting the proposed association of VPS13C and its deletion constructs with Rab7 on the surface of lysosomes. **d**, Fluorescence image of live RPE1 cells expressing mCh-ATG2C_VPS13C labelled with LysoView 640 and LipidSpot 488 (a lipid droplet marker; right). Magnified views of the boxed region are provided (right). **e**, Predicted structure of the ATG2C domain of VPS13C based on AlphaFold3 (left) and high-power views of the amphipathic helices

(right). **f**, Fluorescence images of live RPE1 cells expressing mCh–ATG2C_VPS13C before and after treatment with 1 mM LLOMe (left). Boxed regions are shown at higher magnification demonstrating co-localization of mCh-ATG2C_VPS13C with Lamp1–GFP after LLOMe treatment (right). **g**, Putative model illustrating how binding of the ATG2C_VPS13C domain to the bilayer may release an auto-inhibitory conformation of VPS13C to allow its binding to Rab7 on lysosomal surfaces. **h**, Fluorescence images of live RPE1 cells expressing mCh–ATG2C_VPS13C and GFP–OSBP after treatment with 1 mM LLOMe (left) or 20 nM OSW1 (right). Grey dashed lines in the images represent cell outlines. Insets: magnified views of the boxed regions. **f,h**, Lipid droplets are labelled with asterisks. **b,d,f,h**, The experiments were repeated three times with similar results.

as a brake that prevents the VAB domain from accessing Rab7 in the absence of lysosome damage (Fig. 5c).

To further explore how the ATG2C-PH domain portion of VPS13C controls its binding to lysosomes, we expressed mCh-tagged versions of the ATG2C (mCh–ATG2C$_{VPS13C}$) and PH (mCh–PH) domains separately in RPE1 cells (Fig. 5a,d, and Extended Data Fig. 6a). Under basal conditions, mCh–PH predominantly localized to the cytosol (Extended Data Fig. 6A), whereas mCh–ATG2C$_{VPS13C}$ exhibited strong lipid-droplet binding (Fig. 5d), as reported previously[10]. This binding is consistent with the amphipathic nature of the four parallel helices that constitute this domain, as lipid-droplet binding due to packing defects of the phospholipid monolayer surrounding them is a feature of many amphipathic helices[55,56] (Fig. 5e). Following LLOMe treatment, an additional appearance of an mCh–ATG2C$_{VPS13C}$-positive signal that overlapped with LAMP1 was observed (Fig. 5f), whereas mCh–PH remained primarily cytosolic (Extended Data Fig. 6a). These findings indicate that the ATG2C$_{VPS13C}$ domain can detect damaged lysosome membranes and bind to them.

Together, these results support a model according to which under basal conditions intramolecular interactions involving the C-terminal ATG2C$_{VPS13C}$ domain and/or PH domain prevent the VAB domain from accessing Rab7. Following lysosomal membrane damage, binding of the ATG2C$_{VPS13C}$ domain to these membranes results in a conformational change of the C-terminal moiety of VPS13C, which releases the VAB domain and enables its interaction with Rab7, thereby further promoting and stabilizing the interaction of VPS13C with lysosomes (Fig. 5g).

### The PITT pathway is not involved in VPS13C recruitment

Acute recruitment of VPS13C to lysosomes is not the only event leading to the formation of ER–lysosome tethers in response to lysosome damage. It was reported that exposure to LLOMe results in a rapid accumulation of PI4P on their surface via the recruitment of PI4K2A. PI4P at the lysosome surface in turn recruits ER-anchored oxysterol-binding protein (OSBP)-related proteins (ORP) family members, that is, shuttle-type lipid-transfer proteins that mediate transport of PI4P to the ER in exchange for countertransport of PtdSer and cholesterol from the ER to the lysosomal surface, probably helping repair lysosome damage[32–34]. The timing of this response, referred to as the PITT pathway[32], is similar to the timing of VPS13C recruitment to lysosomes (within minutes after LLOMe treatment), prompting us to investigate whether PI4P contributes to the recruitment of the ATG2C domain of VPS13C. To do so, we evaluated the impact of OSW-1 (ref. 57) on the recruitment of mCh–ATG2C$_{VPS13C}$ to lysosomes in RPE1 cells. OSW1 is a natural compound that blocks the PI4P–cholesterol countertransfer function of OSBP and thus robustly enhances the presence of PI4P on lysosomes[58]. In agreement with previous observations[58–60], we found that GFP–OSBP was exclusively localized in the Golgi complex region under basal conditions (Extended Data Fig. 6b) and a pool of GFP–OSBP rapidly translocated to lysosomes after the cells were treated with OSW-1 (Extended Data Fig. 6b), consistent with PI4P accumulation on their membranes, whereas no obvious translocation of ATG2C$_{VPS13C}$ to these organelles occurred (Fig. 5h and Extended Data Fig. 6b). Both GFP–OSBP and ATG2C$_{VPS13C}$, however, could be recruited to lysosomes in response to LLOMe-induced damage (Fig. 5h and Extended Data Fig. 6c). Thus, ATG2C$_{VPS13C}$ is not a PI4P effector. Similar results were obtained with full-length VPS13C, that is, in VPS13C$^{mClover-Flp-In}$ cells, OSBP translocated to lysosomes in response to OSW-1 but full-length VPS13C did not (Extended Data Fig. 7a), providing further support to the conclusion that the recruitment of VPS13C to damaged lysosomes is independent of PI4P. In addition, although both VPS13C and OSBP were rapidly recruited to lysosomes following LLOMe treatment, the recruitment of VPS13C occurred earlier than the recruitment of OSBP (Extended Data Fig. 7b). We conclude that although both VPS13C and OSBP detect LLOMe-induced lysosome damage, the mechanisms of their recruitment are different and the main trigger for VPS13C binding

to damaged lysosomes is a perturbation of the bilayer of their membranes (Discussion).

Finally, although the recruitment of VPS13C to lysosomes in response to LLOMe slightly preceded the recruitment of OSBP and IST1, we found that both proteins were still efficiently recruited to lysosomes in *VPS13C*-KO A549 cells following LLOMe treatment (Extended Data Fig. 8a–c). This finding indicates that lysosomal recruitment of OSBP and IST1 is independent of VPS13C.

### Loss of VPS13C increases the fragility of lysosomes

The rapid response of VPS13C to lysosomal damage along with its lipid transfer properties suggest that VPS13C may be important to preserve the integrity of lysosomal membranes and to participate in their repair. We have reported previously that *VPS13C*-KO HeLa cells show alterations of lysosome homeostasis, as revealed by an increased number of lysosomes and activation of the cGAS–STING pathway of innate immunity, possibly reflecting defective and leaky lysosomes. Consistent with these results, immunofluorescence analysis of another cell type, A549 cells, revealed elevated levels of LAMP1 in *VPS13C*-KO cells compared with WT cells (Fig. 6a and Extended Data Fig. 9a). Moreover, fluorescence intensity analysis of cells incubated with LysoView 633, a probe whose fluorescence correlates inversely with lysosomal pH, revealed decreased acidification in *VPS13C*-KO cells (Fig. 6b and Extended Data Fig. 9b).

We used the Gal3 fluorescence assay to directly investigate the impact of VPS13C in the response to acute lysosome damage. To account for the higher number of lysosomes observed in *VPS13C*-KO cells, all Gal3 fluorescence values were normalized to the LAMP1 immunofluorescence. Under basal conditions, immunofluorescence for endogenous Gal3 did not reveal accumulation of this protein in either WT or *VPS13C*-KO cells (Fig. 6c,d, time 0). Following LLOMe treatment, a progressive appearance of Gal3 puncta was observed, which occurred earlier in *VPS13C*-KO cells compared with WT cells (Fig. 6c,d). Similar results were observed when the fluorescence of transiently expressed Gal3–GFP in HeLa cells was monitored (Extended Data Fig. 9c).

We conclude that the loss of VPS13C increases lysosomal fragility, underscoring its important role in resilience of the lysosomal membranes to damage.

### Different kinetics of VPS13C and LRRK2 recruitment

LRRK2 can also bind to stressed/damaged lysosomes[25,26,61,62]. In view of the link of both LRRK2 and VPS13C to PD, we investigated the temporal and mechanistic relationships between VPS13C and LRRK2 recruitment. As various perturbations that induce LRRK2 recruitment and activation (including LLOMe) do so by triggering the conjugation of ATG8 to single membranes (CASM) process[37,63], we focused on agents that regulate this pathway (Fig. 7a). Our investigations revealed that other experimental manipulations that induce CASM, such as chloroquine (a lysosomotropic reagent) or nigericin (an H$^+$/K$^+$ ionophore), which disrupt the acidic luminal pH of lysosomes and thus may have an indirect effect on the lysosomal membrane, also recruit VPS13C (Fig. 7b and Extended Data Fig. 10a). Recruitment of VPS13C to lysosomes in response to nigericin was also recently reported in a mass spectrometry analysis of purified lysosomes[64]. However, saliphenylhalamide (Salip), an agent that increases lysosomal pH by locking the proton pump in an inactive state and is a well-characterized CASM and LRRK2 activator[37,65–67], failed to recruit VPS13C (Fig. 7c(top)), although it induced a loss of LysoView 633 (a pH indicator) fluorescence (Fig. 7c(bottom)), proving its effectiveness in elevating lysosomal pH. Furthermore, overexpression of SopF, a *Salmonella* effector protein that blocks the interaction between proton pump and ATG16L1 (refs. 65,68) and is a potent inhibitor of CASM, did not prevent the recruitment of VPS13C by LLOMe (Fig. 7d). Thus, although both VPS13C and LRRK2 are recruited to lysosomes in response to their perturbation, VPS13C recruitment, in contrast to LRRK2, is not dependent on CASM.

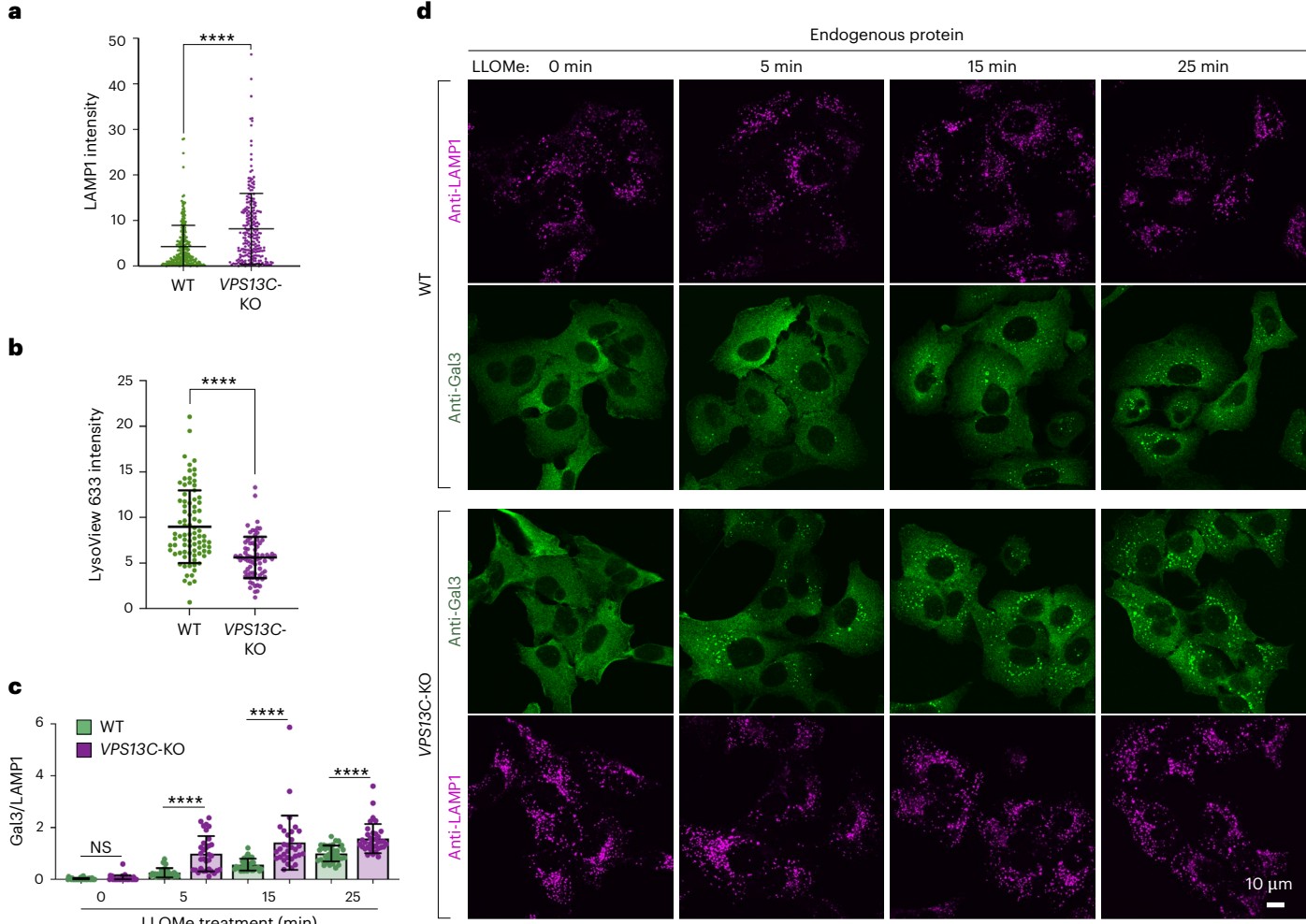

**Fig. 6 | Loss of VPS13C increases the fragility of lysosomes. a,b,** Intensity of punctate LAMP1 immunofluorescence (**a**) and LysoView 633 fluorescence (a pH-sensitive lysosome probe; **b**) per WT and *VPS13C*-KO A549 cell. **a**, *n* = 240 cells per group. **b**, *n* = 84 WT and 76 *VPS13C*-KO cells. **c,d,** Gal3-to-LAMP1 ratio (**c**) determined from the images in **d. c,** Normalized fluorescence relative to WT cells treated with LLOMe for 25 min. NS, *P* = 0.2886; *n* = 32 (WT, 0 min; WT, 5 min; and *VPS13C*-KO, 5 min), 33 (*VPS13C*-KO, 0 min), 34 (WT, 15 min), 31 (*VPS13C*-KO,

15 min), 39 (WT, 25 min) and 35 (*VPS13C*-KO, 25 min) images analysing >300 cells. **d,** The images depict fluorescence microscopy of WT and *VPS13C*-KO A549 cells immunolabelled with antibodies to Gal3 and LAMP1 following treatment with or without 1 mM LLOMe for the indicated times. Fluorescence images from a representative experiment. **a–c,** Three biological replicates. Data are the mean ± s.d. Data were compared using a two-sided Student's *t*-test; ****P* < 0.0001; NS, not significant. Numerical data are provided.

We also directly compared the dynamics of VPS13C and LRRK2 recruitment induced by LLOMe. We conducted these experiments in HeLa cells, which show robust LLOMe-induced lysosomal recruitment of LRRK2 compared with the VPS13C^mClover-Flp-In cell line used in most of our experiments. Both VPS13C^Halo and GFP–LRRK2 predominantly localized in the cytosol, with VPS13C showing some lysosomal binding under basal conditions. VPS13C was recruited to lysosomes within minutes following LLOMe treatment (Fig. 7e), consistent with observations in VPS13C^mClover-Flp-In (Fig. 1b) and RPE1 cells (Fig. 4d,e). Notably, in contrast to what we had observed in VPS13C^mClover-Flp-In and RPE1 cells, where VPS13C accumulation at endolysosomes persisted for at least 30 min after LLOMe addition, VPS13C recruitment in HeLa cells was transient, peaking at about 10 min after the addition of LLOMe (Fig. 7e and Extended Data Fig. 10b). Importantly, the recruitment of LRRK2 was much delayed, becoming detectable approximately 1 h after the addition of LLOMe when VPS13C had mostly dissociated from lysosomes in these cells (Fig. 7e), and involved a smaller number of lysosomes. A similar sequential recruitment of VPS13C and LRRK2, with fewer lysosomes becoming positive for LRRK2, was also observed in RPE1 cells following LLOMe treatment (Extended Data Fig. 10c).

Together, these results demonstrate that although both VPS13C and LRRK2 respond to lysosome damage, the mechanisms controlling their recruitment differ significantly.

## Discussion

Our study reveals that recruitment of VPS13C is an early response to endolysosome damage, possibly to provide a path for bulk lipid transport from the ER to their membranes. This suggests that lipid transport mediated by this protein via a bridge-like mechanism may play a role in preventing or repairing the rupture of the lysosomal membrane. For this reason, loss-of-function mutations in *VPS13C* may contribute to the development of PD.

These findings are convergent with the recent report that lysosome damage induces the formation/expansion of ER–lysosome contacts mediated by ER-anchored OSBP and ORPs in a process that is dependent on the damage-triggered accumulation of PI4P on the lysosome membrane, the so-called PITT pathway[32–34]. OSBP and ORPs are shuttle-like lipid-transport proteins that bind the ER protein VAP via an FFAT motif and PI4P-rich Golgi–endosomal membranes via a PH domain[69,70]. The PITT pathway, which is activated with a similar time course as the recruitment of VPS13C, is also thought to help provide

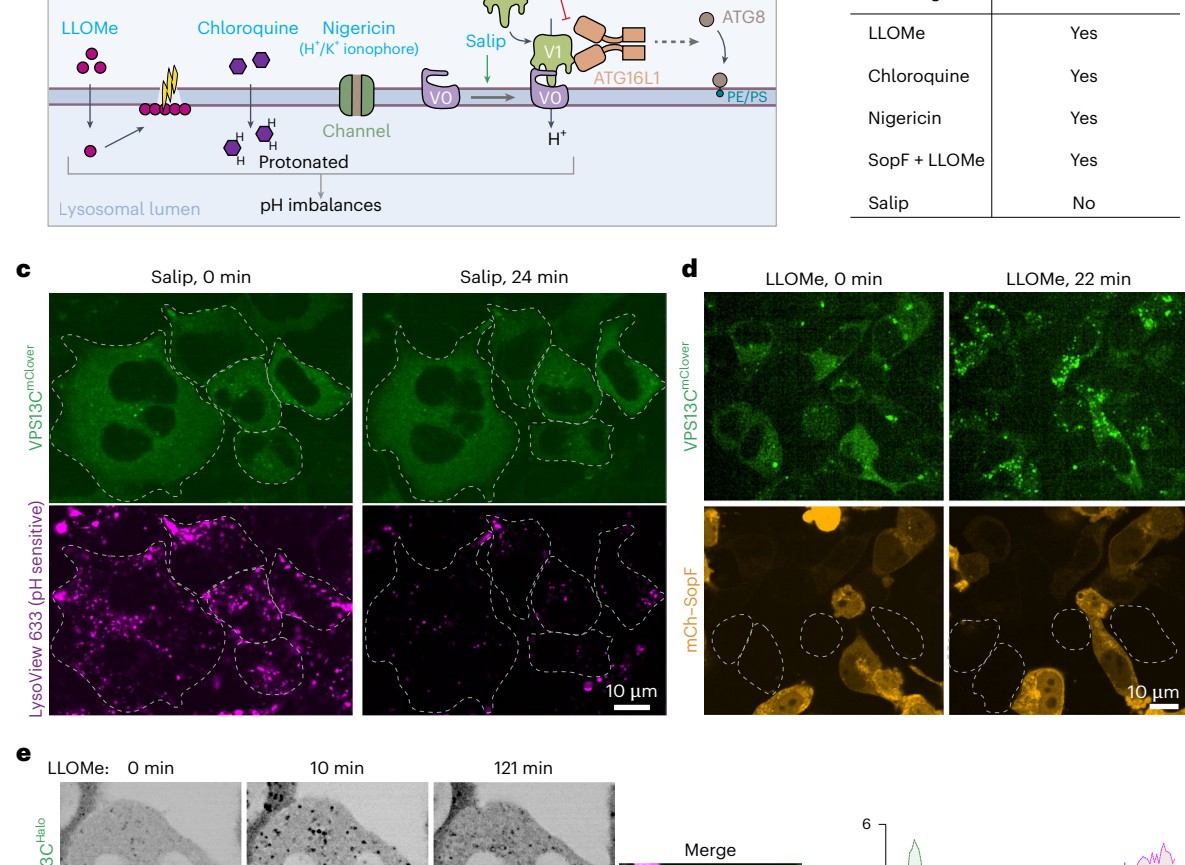

**Fig. 7 | Relationship between VPS13C and LRRK2 recruitment to damaged lysosomes. a**, Schematic illustration of the CASM pathway and experimental manipulations that trigger (blue) or inhibit (red) it. **b**, Summary of agents tested and their impact on the recruitment of VPS13C to lysosomes. **c**, Fluorescence images of live VPS13C$^{mClover-Flp-In}$ cells, labelled with LysoView 633, before and after Salip treatment. Dashed lines represent cell outlines. **d**, Fluorescence images of live VPS13C$^{mClover-Flp-In}$ cells with exogenous mCh–SopF co-expression before and after treatment with 1 mM LLOMe. Cells not expressing mCh–SopF are outlined by dashed lines. **e**, Time-series fluorescence images of live HeLa cells co-expressing VPS13C$^{Halo}$ and GFP–LRRK2 showing recruitment of VPS13C and LRRK2 to lysosomes following treatment with 1 mM LLOMe (left). Individual channel images are shown as inverted greys. Relative punctate fluorescence intensity of VPS13C$^{Halo}$ and GFP–LRRK2 per cell after LLOMe treatment in a single experiment (right). Mean ± s.d. represented by the solid line and shaded area, respectively; *n* = 15 VPS13C and 11 LRRK2 cells were analysed. **c–e**, The experiments were repeated three times with similar results. Further examples in Extended Data Fig. 10b,c. Numerical source data are provided.

phospholipids (specifically PtdSer and cholesterol) to the lysosomal membrane from the ER, but in exchange with the countertransport of PI4P from the lysosome to the ER[69], thus not resulting in net lipid transfer to endolysosomes. Although we have found that the recruitment to lysosomes of VPS13C is independent of PI4P and thus of the PITT pathway, the two processes may synergize given that (1) once an ER–lysosome contact has formed, the population of these contacts by other ER–lysosome tethers will be facilitated and (2) the role of OSBP and ORP proteins in regulating the lipid composition of lysosome membranes may be complementary to the expected role of VPS13C in mediating net lipid flow to such membranes.

Activation of the PITT pathway in response to lysosome damage was reported to be followed by the recruitment at ER–lysosome

contacts of ATG2A, another bridge-like lipid-transport protein[32]. The ORP family-dependent PtdSer accumulation on the lysosome membrane and/or CASM were proposed to be responsible for such recruitment. However, our results indicate that the recruitment of VPS13C represents a distinct response to lysosome damage that occurs independently of the PITT and CASM pathways[11,71,72].

Other proteins that accumulate with rapid kinetics at the lysosome surface after damage are ESCRT complex components[27–29]. Such recruitment was proposed to help mediate lysosome-membrane repair, as the ESCRT mediates the formation of spiral filaments that seal membrane holes in a variety of cellular contexts[31,73–77]. Thus, the delivery of membrane lipids may function cooperatively with ESCRT in membrane resilience or repair. It is also of interest that one of the

proposed functions of the single yeast VPS13 protein is to provide phospholipids to endolysosomes for the generation of intraluminal vesicles, an ESCRT-dependent process[78]. Whether such a function applies to mammalian VPS13C and, if so, how such a function is related to the recruitment of VPS13 in response to lysosome stress remains to be elucidated.

Although VPS13C recruitment requires Rab7, which binds the VAB domain of VPS13C, enhanced Rab7 activity at the lysosomes does not seem to be responsible for such recruitment, as recruitment of the VAB-domain-only construct is not increased by LLOMe. We have also ruled out Rab7 phosphorylation, PI4P and $Ca^{2+}$ as the signal that causes VPS13C recruitment. We have found instead that such signal is a perturbation of the lysosomal membrane that triggers binding of the $ATG2C_{VPS13C}$ domain, most probably resulting in the release of an auto-inhibited conformation of VPS13C that prevents its access to Rab7. Both the VAB-domain-only and VPS13C constructs that contain this domain but lack the ATG2C and PH domains bind efficiently to endolysosomes also under basal conditions. In contrast, full-length VPS13C and its C-terminal fragment ($VPS13C_{C-ter}$), which contain both the VAB domain and the ATG2C-PH region, do not bind Rab7 in basal conditions. A potential clue about the mechanism that mediates the binding of the ATG2C domain of VPS13C to a damaged membrane comes from its property of binding lipid droplets via its amphipathic helices[10]. Many proteins that bind lipid droplets do so by recognizing packing defects present in the monolayer that surrounds them[55,56]. Bilayer packing defects generated by lysosome damage may be the signal that triggers $ATG2C_{VPS13C}$ recruitment. Providing support for this possibility, it was shown that SPG20, another protein that contains amphipathic helices, is also recruited to lysosomes within minutes following damage of their membranes[79,80]. Interestingly, the lysosomal recruitment of SPG20, like that of VPS13C, is independent of PI4P and $Ca^{2+}$ signalling[79].

Finally, although both VPS13C and LRRK2 are responsive to lysosomal stress/damage, our study reveals distinct mechanisms and kinetics in their recruitment. VPS13C exhibits rapid recruitment, suggesting an early response in membrane repair/protection, whereas LRRK2 recruitment, which is linked to the activation of CASM, is delayed, pointing to its involvement in late-stage processes, potentially resulting from severe lysosomal damage. Moreover, LRRK2 is recruited to a smaller number of lysosomes. However, a function of both proteins in a response to lysosome damage strengthens evidence[20–22] that impairments of such a response in vulnerable cell types are critical contributors to the development of PD.

## Online content

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

## Methods

### DNA plasmids
A plasmid containing codon-optimized complementary DNA encoding human VPS13C with an internal Halo protein after amino acid residue 1914 (VPS13C[Halo]; Research Resource Identifier (RRID): Addgene 232864) was generated by and purchased from GenScript Biotech. Plasmids lacking the C-terminal ATG2C and PH domains of VPS13C, designated as VPS13C-Δ(ATG2C-PH)[Halo] (RRID: Addgene 232867), as well as those expressing the ATG2C or PH domain of VPS13C (mCh–ATG2C[VPS13C] (RRID: Addgene 232868) and mCh–PH (RRID: Addgene 232869)) were generated and purchased from Epoch Life Science, Inc. Halo-SspB-PPM1H (RRID: Addgene 232870) was constructed as follows. First, the Halo tag was amplified by PCR from the VPS13C[Halo] plasmid. Subsequently, the SspB peptide sequence was added by overlapping PCR and the construct was ligated into the 3×Flag-CMV10 vector using InFusion cloning employing the SacI and NotI cut sites. Next, PPM1H was amplified by PCR from the HA–PPM1H (MRC Reagents and Services, DU62789) plasmid and inserted downstream of SspB using InFusion cloning with the NotI and BamHI cut sites. The following plasmids were previously generated in our laboratories: VPS13C[mClover3] (RRID: Addgene 118760), mCh–VAB (RRID: Addgene 232865), mCh–VPS13C[C-ter] (RRID: Addgene 232866), mCh–VAPB (RRID: Addgene 108126), mCh–OSBP (RRID: Addgene 232871), GFP–LRRK2 (RRID: Addgene 232872), HA–PPM1H (MRC Reagents and Services, DU62789), GFP–LRRK1 (MRC Reagents and Services, DU30382), GFP–LRRK1[K746G] (MRC Reagents and Services, DU67083) and GFP–LRRK1[D1409A] (MRC Reagents and Services, DU67084). The HA–PPM1H and GFP–LRRK1 constructs were further subcloned into the pmScarlet vector, between XhoI and BamHI, using InFusion cloning. The following constructs were obtained from Addgene: mCh–SopF from L. Knodler (RRID: Addgene 135174), LAMP1-mCh–iLID from L. Kapitein (RRID: Addgene 174625), Lamp1–RFP from W. Mothes (RRID: Addgene 1817) and mCh–Gal3 from H. Meyer (RRID: Addgene 85662). IST1–Apple was a gift from P. Hanson (University of Michigan School of Medicine, Ann Arbor, MI).

### Antibodies and reagents
**Antibodies.** The following antibodies were used at the indicated dilutions: anti-lamp1 (Cell Signaling Technology, 9091; RRID: AB_2687579; for western blotting (WB), 1:2,000), anti-LAMP1 (Abcam, ab25630; RRID: AB_470708; for immunofluorescence (IF), 1:100), anti-Gal3 (R&D Systems, IC1154G; RRID: AB_10890949; for IF, 1:50), anti-VPS13C (Proteintech, 29844-1-AP; RRID: AB_3086177; for WB, 1:1,000), anti-GM130 (BD Biosciences, 610822; RRID: AB_398141; for WB, 1:2,000), anti-PDI (CST, 2446S; RRID: AB_2298935; for WB, 1:1,000), anti-VAPB (Sigma-Aldrich, HPA013144; RRID: AB_1858717; for WB, 1:4,000), anti-Rab7 (Cell Signaling Technology, 9367; RRID: AB_1904103; for WB, 1:1,000), mouse monoclonal anti-Rab7A (Sigma-Aldrich, R8779; RRID: AB_609910; for WB, 1:2,000), anti-pSer72 Rab7 (Abcam, ab302494; RRID: AB_2933985; for WB, 1:1,000), anti-GAPDH (Proteus, 40-1246; for WB, 1:1,000), anti-GFP (Abcam, ab290; RRID: AB_303395; for WB, 1:1,000), anti-tubulin (Sigma-Aldrich, T5168; RRID: AB_477579; for WB, 1:2,000), anti-mCh (Abcam, Ab125096; RRID: AB_11133266; for WB, 1:1,000), anti-LRRK1 (MRC Reagents and Services, S405C; for WB, 1 μg ml⁻¹), anti-IKKε (Cell Signaling Technology, 3416S; for WB, 1:2,000), anti-TBK1 (Cell Signaling Technology, 3504S; RRID: AB_2255663; for WB, 1:2,000) and anti-pSer172 TBK1 (Cell Signaling Technology, 5483S; RRID: AB_10693472; for WB, 1:1,000). The secondary antibodies IRdye 800CW (926-32213) and IRdye 680LT (926-68020) for WB (1:10,000) were from LICORbio.

**Reagents and final concentrations.** The following reagents were used at the indicated concentrations: 1 mM LLOMe (Sigma-Aldrich, L7393-500MG; Chemical Abstracts Service (CAS) number: 16689-14-8), 200 μM chloroquine (Sigma-Aldrich, C6628; CAS: 50-63-5), 10 μM nigericin (Sigma-Aldrich, N7143; CAS: 28643-80-3), 2.5 μM Salip

(Omm Scientific), 20 nM OSW-1 (MedChem Express, HY-101213; CAS: 145075-81-6), LysoView 633 (Biotium, 70058; 1:2,000), LysoView 640 (Biotium, 70085; 1:2,000), LipidSpot 488 (Biotium, 70065; 1:2,000), 100 nM human RAB7A siRNA (Horizon Biosciences, M-010388-00-0005), 0.1 μg ml⁻¹ tetracycline (Thermo Scientific, A39246; CAS: 64-75-5), 200 μM E64d (Cayman Chemical, 13533; CAS: 88321-09-9), Lipofectamine RNAiMAX (Thermo Scientific, 13778030; 1:3) and FuGene HD (Promega, E2311; 1:3).

### Generation of tetracycline-inducible VPS13C[mClover-Flp-In] stable cell line
The Flp-In TREx 293 cell line (Invitrogen, R78007), which contains a single Flp recombination target (FRT) site in its genome, was maintained in Dulbecco's modified Eagle's medium (DMEM) containing 10% fetal bovine serum (FBS), 200 μg ml⁻¹ zeocin and 15 μg ml⁻¹ blasticidin S. To establish the tetracycline-inducible cell line, pcDNA5/FRT/TO vector containing the cDNA sequence of VPS13C[mClover] and pOG44 vector (Invitrogen), which expresses Flp recombinase, were co-transfected into Flp-In TREx 293 cells using FuGene HD (Promega) according to the manufacturer's recommendations. As a negative control, pcDNA5/FRT (RRID: Addgene 41000) vector only was used for transfection. The cells were washed 24 h post transfection and fresh medium containing blasticidin but not zeocin was added to the cells. At 48 h post transfection, 200 μg ml⁻¹ hygromycin B (Invitrogen) was added to the culture for selection. The medium was changed every 3–4 days until hygromycin-resistant colonies were evident. Expression of the protein was induced by the addition of 0.1 μg ml⁻¹ tetracycline in the medium for 24 h and checked on SDS–PAGE gels, followed by western blot analysis. A detailed method for the generation of the cells (RRID: CVCL_E6IS) is available[81].

### Generation of *VPS13C*-KO A549 cell line
A549 (RRID: CVCL_0023) cells were transfected with 1.5 μg PX459 plasmid (plasmid number 62988, Addgene) containing a single guide RNA[15] targeting *VPS13C* using Lipofectamine 2000 (Thermo Fisher Scientific). The cells were selected in complete DMEM containing 1 μg ml⁻¹ puromycin 24 h after transfection. The medium was replaced with fresh puromycin-containing medium 48 and 72 h after transfection. After three days of puromycin selection, single clones were obtained using serial dilution and then screened by western blotting. Cells found to be lacking VPS13C (RRID: CVCL_E6IP) were selected. A detailed method for the generation of the cells is available at https://doi.org/10.17504/protocols.io.eq2lynx5wvx9/v1.

### Generation and culture of MEFs
Wild-type and homozygous *LRRK1*-KO MEFs[49] were isolated from littermate-matched mouse embryos resulting from crosses between heterozygous *LRRK1*-KO/WT mice on embryonic day E12.5 (E12.5) as described[82].

MEFs generated from double-KO mice that do not express TBK1 and IKKε were a gift from S. Akira (Department of Host Defense, University of Osaka, Japan) and were described previously[83]. Genotypes were verified via allelic sequencing and immunoblot analysis. Cells were cultured in DMEM medium containing 10% (vol/vol) FBS, 2 mM ʟ-glutamine, 100 U ml⁻¹ penicillin–streptomycin, 1 mM sodium pyruvate and 1×non-essential amino acid solution (Life Technologies, Gibco). The cells were regularly tested for mycoplasma PCR products using a Lonza Mycoplasma kit.

Mice were maintained at the University of Dundee under specific-pathogen-free conditions. All animal studies were ethically reviewed and carried out in accordance with Animals (Scientific Procedures) Act 1986 and regulations set by the University of Dundee and the UK Home Office. Animal studies and breeding were approved by the University of Dundee ethical committee and performed under a UK Home Office project licence. The mice were housed at ambient

temperature (20–24 °C) and humidity (45–55%), and maintained on a 12 h light–12 h dark cycle with free access to food (SDS RM no. 3 autoclavable) and water.

## Cell culture and transfection

VPS13C$^{mClover-Flp-In}$ and HeLa cells (catalogue number RCB5388; RRID: CVCL_R965) were cultured at 37 °C with 5% $CO_2$ in DMEM medium supplemented with 10% FBS. RPE1 cells (catalogue number CRL-4000; RRID: CVCL_4388) were cultured at 37 °C with 5% $CO_2$ in DMEM/F12 medium supplemented with 10% FBS. For live-cell imaging experiments, the cells were seeded onto glass-bottomed dishes (MatTek). For biochemical experiments, the cells were plated in 6-cm-diameter dishes. Following incubation for 24 h, the cells were transfected using FuGene HD (Promega) according to the manufacturer's recommendations. Tetracycline (0.1 ug ml$^{-1}$ working concentration) was concurrently added to the medium to induce VPS13C$^{mClover}$ expression and the cells were imaged 20–24 h later. Transfection of siRNA was carried out using Lipofectamine RNAiMax (Thermo Fisher Scientific) with 100 nM siRNA pool as per the manufacturer's recommendations. The cells were either imaged or collected for western blot analysis 48 h later.

## Live-cell imaging

Growth media were changed with live-cell imaging solution (Life Technologies) shortly before imaging. Imaging was performed at 37 °C in 5% $CO_2$ using a Nikon Ti2-E inverted microscope equipped with a Spinning Disk Super Resolution by Optical Pixel Reassignment Microscope (Yokogawa CSU-W1 SoRa, Nikon) and Microlens-enhanced Nipkow Disk with pinholes and a ×60 SR Plan Apo IR oil-immersion objective. For the LysoView experiments, the cells were incubated in complete DMEM for 30 min, washed twice with fresh medium and then imaged in live-cell imaging solution. For the LLOMe experiments, a 2× stock solution was prepared in the imaging solution and added to the dish during imaging.

For the light-dependent recruitment of PPM1H to lysosomes, the same Yokogawa CSU-W1 SoRa microscope was used. Recruitment was achieved with 200 ms pulses of the 488 nm laser. Imaging was performed at 37 °C in 5% $CO_2$, with a ×60 SR Plan Apo IR oil-immersion objective.

A detailed description of cell culture, transfection and imaging is available[84].

## Immunofluorescence

Wild-type or *VPS13C*-KO A549 cells cultured on glass coverslips were fixed with 4% paraformaldehyde in PBS (Gibco, 14190144) for 20 min at room temperature, washed three times with PBS, permeabilized with 1×Tris-buffered saline with Tween buffer (Santa Cruz Biotechnology, sc-281695) for 30 min at room temperature and blocked with filtered PBS containing 3% (wt/vol) BSA for 1 h at room temperature. The coverslips were then incubated overnight with antibodies (anti-LAMP1 (H4A3); Abcam, ab25630, 1:100; and anti-Gal3 Alexa Fluor 488-conjugated; R&D Systems, IC1154G; 1:50) at 4 °C, followed by three washes in PBS. The samples were then incubated with secondary antibodies (1:1,000; Alexa Fluor 546, Invitrogen) in PBS containing 3% BSA for 1 h at room temperature in the dark and washed three times with PBS. Finally, the coverslips were mounted onto slides. A detailed method is available at https://doi.org/10.17504/protocols. io.14egn741mv5d/v1.

## Generation of SPIONs

An established protocol was used to generate SPIONs[15,37,85]. Briefly, 10 ml of 1.2 M $FeCl_2$ (Sigma-Aldrich, 220299) and 10 ml of 1.8 M $FeCl_3$ (Sigma-Aldrich, 157740) were combined slowly by stirring. Next, 10 ml of 30% $NH_4OH$ (Sigma-Aldrich, 320145) was slowly added with stirring for 5 min. The resulting particles were then washed three times with 100 ml water. The particles were resuspended in 80 ml of 0.3 M HCl and stirred for 30 min, followed by the addition of 4 g dextran

(Sigma-Aldrich, D1662) and stirring for 30 min. The particles were transferred into dialysis tubing (Thermo Fisher, 68100) and dialysed with double-distilled water for at least two days with multiple water changes. The particles were centrifuged at 26,900$g$ for 20 min to remove large aggregates and stored at 4 °C until further use. The detailed protocol is available[86].

## Isolation of lysosomes using SPIONs

Cells were cultured in 15 cm dishes until they reached 90–95% confluency. The medium was then removed and the cells were washed once with PBS (Thermo Fisher, 10010023). Subsequently, 20 ml fresh DMEM medium (supplemented with FBS and penicillin–streptomycin) containing 10% SPIONs particles and 10 mM HEPES pH 7.4 (Thermo Fisher, 15630080) was added to the cells, which were then incubated at 37 °C for 24 h. The medium containing particles was removed and the cells were washed once with PBS. The cells were trypsinized (Thermo Fisher, 25200056), re-plated evenly on four 10 cm culture dishes and incubated at 37 °C for 24 h. Following the removal of medium, the cells were washed three times with PBS. Next, the cells were collected in PBS by scraping on ice, followed by centrifugation at 1,000 rpm and 4 °C for 5 min. The cell pellet from each dish was resuspended in 1 ml homogenization buffer (HB) containing 5 mM Tris (pH 7.4; RPI, T60040), 250 mM sucrose (Sigma, S0389), 1 mM EGTA (Sigma-Aldrich, E4378) and phosphatase/protease inhibitors (PhosSTOP; Roche, 4906837001; and cOmplete mini EDTA-free; Roche, 11836170001). Whole-cell lysate was generated using a Dounce homogenizer (DWK Life Sciences, 357538), followed by centrifugation at 800$g$ and 4 °C for 10 min to collect the supernatant (whole-cell lysate). The whole-cell lysate was loaded onto a LS column (Miltenyi Biotec, 130042401), and the flow-through was collected and reapplied to the same column. After washing the column once with 3 ml HB, the column was removed from the magnetic stand and the bound fraction was eluted with 2.5 ml HB. The bound fraction was then centrifuged at 55,000 rpm and 4 °C for 1 h to generate the lysosomal pellet, which was resuspended in 50 µl HB and stored at −20 °C until further processing. A detailed description of the protocol is available[87].

## Western blotting

Cultured cells were lysed on ice through repeated pipetting in 2% SDS supplemented with protease inhibitor cocktail (Roche) and PhosStop phosphatase inhibitor (Roche). Each sample was further sonicated for 30 s to break the DNA and reduce the viscosity of the samples. Total protein was then measured using a Pierce BCA assay (Thermo Fisher Scientific). The samples were then prepared for western blotting: equal protein concentrations were mixed with SDS loading buffer (final concentration of 0.05% bromophenol blue, 0.1 M dithiothreitol, 10% glycerol, and 2% SDS in 0.05 M Tris–HCl pH 6.8) and incubated at 95 °C for 5 min. Proteins were separated on Mini PROTEAN TGX 4–20% Tris–glycine gels (Bio-Rad) before transfer to nitrocellulose membranes for 2 h at 100 V and 4 °C in transfer buffer containing 25 mM Tris, 192 mM glycine and 20% methanol in milliQ purified water. The membranes were blocked with 5% milk in Tris-buffered saline containing 0.1% Tween 20 (TBST) for 1 h. The membranes were then incubated overnight at 4 °C with primary antibodies in 3% BSA in PBST. The next day the membranes were washed three times in TBST and then incubated (room temperature for 1 h) with secondary antibodies conjugated to IRdye 800CW or IRdye 680LT (Licor; 1:10,000) in 5% milk in TBST, washed three times in TBST and finally imaged using an Odyssey imaging system (LI-COR) according to the manufacturer's protocol. For the experiments shown in Extended Data Fig. 4, cells were lysed and the whole-cell lysates were analysed by immunoblotting as described[88].

## Image analysis

The fluorescence images presented in this study were processed using the Fiji software (RRID: SCR_002285). Quantification of fluorescence

puncta (endolysosomes) or diffuse cytosolic fluorescence was performed as follows. First, the images were converted to eight-bit files and separated into individual channels. Subsequently, an automatic alignment of cells was conducted using the StackReg plugin (National Institute of Health) in Fiji. Next, outlines delineating individual cells were manually drawn and adjustments of image thresholds were manually performed in each channel to ensure consistency across all images within the same channel. To isolate the relative puncta signal, the threshold was adjusted to eliminate all cytosolic signals by elevating the minimum value. Conversely, for the relative cytosolic signal, the threshold was adjusted to remove all puncta signals by reducing the maximum value. The average intensity of each cell over time was calculated using the Time Series Analyzer V3 plugin (RRID: SCR_014269) in Fiji. To obtain the final relative intensity over time, the values from each cell were normalized by subtracting the initial values, thus setting the first value as zero. For quantifying the puncta intensity ratio between the Gal3 and LAMP1 fluorescent channels, we first needed to differentiate the true fluorescent signal from noise and background (Cytosol).

For quantification of the Gal3-to-LAMP1 fluorescence ratios, we used the Trainable Weka Segmentation plugin in Fiji and user interaction to train a classifier for each channel. Three datasets were used for training the classifiers, one for each channel. They were then applied to all datasets to mask the Gal3 and LAMP1-positive regions in the corresponding channel. Within those masked regions, measurements of integrated density using Fiji's measure tool were performed for each fluorescence channel, after which a ratio between these two measurements was obtained for each dataset using a script that is available for download (https://doi.org/10.5281/zenodo.14814757; ref. 89).

### Statistics and reproducibility
All of the data are based on multiple experiments with independent biological replicates. The number of data points in each experiment is specified in the figure legends. No statistical methods were used to pre-determine sample sizes but our sample sizes are similar to those reported in previous publications. For fluorescence microscopy image acquisition, cells were selected from random microscopy fields to avoid artefacts and to improve the robustness of data collection. The investigators were not blinded to the conditions of the experiments during data analysis. No samples or data points were excluded. The data distribution was assumed to be normal but this was not formally tested. GraphPad Prism 8 software (v8.0.1; http://www.graphpad.com/; RRID: SCR_002798) was utilized for all statistical analyses.

### Reporting summary
Further information on research design is available in the Nature Portfolio Reporting Summary linked to this article.

### Data availability
Supporting data are provided with this paper and are also available at *Zenodo* (https://doi.org/10.5281/zenodo.14845889)[90]. The protocols and key laboratory materials used and generated in this study are listed with their persistent identifiers in Supplementary Table 1, which is also available at *Zenodo* (https://doi.org/10.5281/zenodo.14846055)[91]. An earlier version of this manuscript was posted on *bioRxiv* on 8 June 2024 (ref. 92). Source data are provided with this paper.

### Code availability
The script for image quantification is available at *Zenodo* (https://doi.org/10.5281/zenodo.14814757)[89].

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

## Acknowledgements
We thank M. Hanna for advice and discussion, and P. Hanson for the gift of the IST1 clone. The *TBK1* and *IKKε* double-KO MEFs were a gift from S. Akira (Department of Host Defense, University of Osaka, Japan). This work was supported in part by grants from the NIH (grant numbers DA018343 and NS36251 to P.D.C., and GM105718 to S.M.F.), Parkinson's Foundation (grant number PF-RCE-1946 to P.D.C and S.M.F.), UK Medical Research Council (grant number MC_UU_00018/1 to D.R.A.) and Aligning Science Across Parkinson's grants through the Michael J. Fox Foundation for Parkinson's Research (grant numbers ASAP-000580 to P.D.C. and S.M.F., and ASAP-000463 to D.R.A). For the purpose of open access, the authors have applied a CC BY public copyright licence to all author accepted manuscripts arising from this submission.

## Author contributions
Conceptualization: X.W., P.D.C., S.M.F. and D.R.A. Investigation: X.W., P.X., A.B.-D., W.H.-C., S.C., B.T.J., F.T., G.T. and L.S. Supervision: P.D.C., S.M.F. and D.R.A. Writing (original draft): X.W. and P.D.C. Writing (review and editing): X.W., P.D.C., S.M.F. and D.R.A.

## Competing interests
P.D.C. is a member of the Scientific Advisory Board of Casma Therapeutics. The other authors declare no competing interests.

## Additional information
**Extended data** is available for this paper at https://doi.org/10.1038/s41556-025-01653-6.

**Correspondence and requests for materials** should be addressed to Pietro De Camilli.

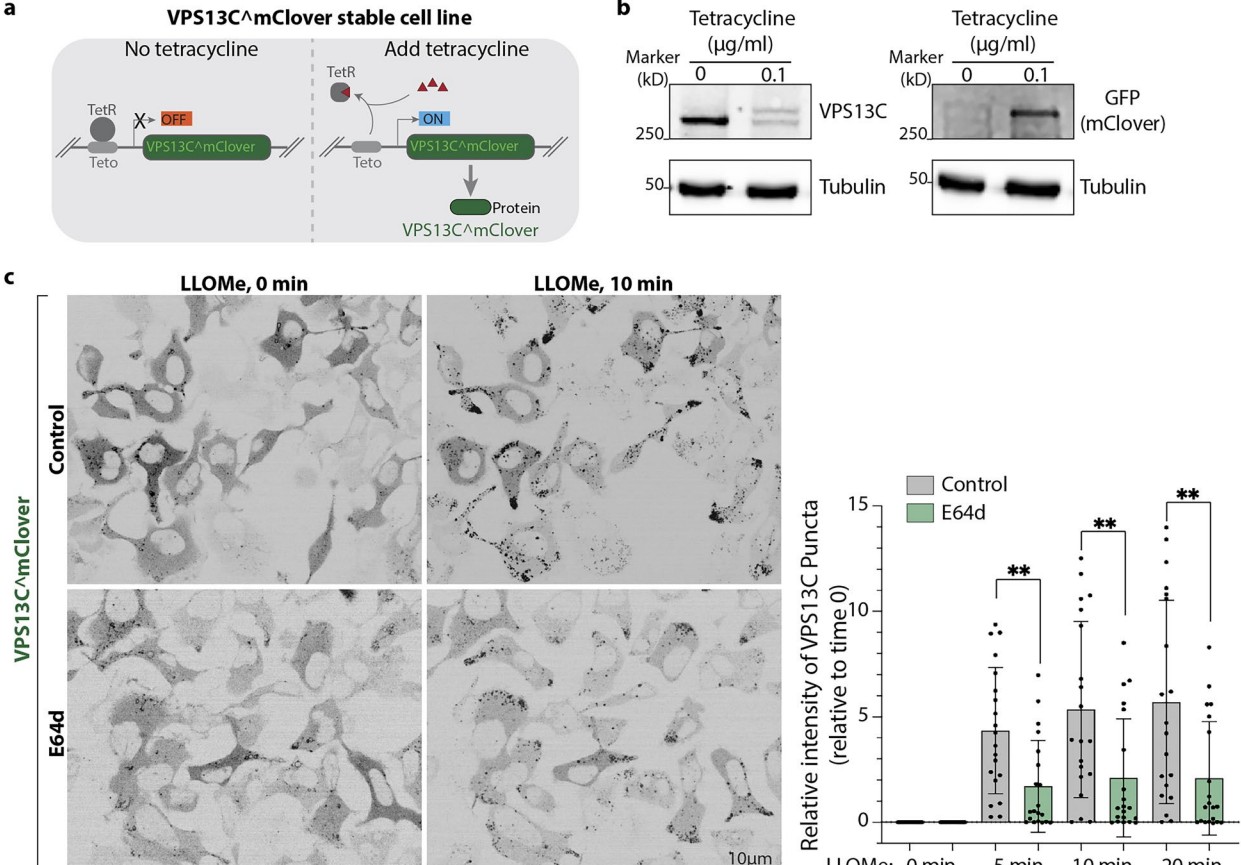

**Extended Data Fig. 1 | Validation of the VPS13C^mClover-Flp-In cell line under the control of tetracycline and effect of the cathepsin C inhibitor E64D on VPS13C recruitment to endolysosomes. a**, Schematic drawing showing the experimental system used for the inducible stable expression of VPS13C^mClover. **b**, Western blots for the indicated proteins of whole cell lysates from VPS13C^mClover-Flp-In cells under the control of tetracycline, with or without tetracycline (0.1 μg/ml) treatment for 24 hours. Tubulin was used as a loading control. The experiment was repeated three times with similar results. **c**, Live fluorescence images of VPS13C^mClover-Flp-In cells before and after a

10 min exposure to 1 mM LLOMe with and without the additional presence of the cathepsin inhibitor E64d (200 μM). The experiment was repeated three times with similar results. Quantification of the intensity of the VPS13C^mClover punctate fluorescence per cell from a representative experiment is shown on the right. n = 20 cells (Control or E64D) were analysed. Graph shows the normalized fluorescence relative to time 0. Data were compared using two-sided $t$-tests. Error bars represent ±SD. **represent P = 0.0029, P = 0.0065 and P = 0.0057, respectively. Numerical data and unprocessed blots are provided.

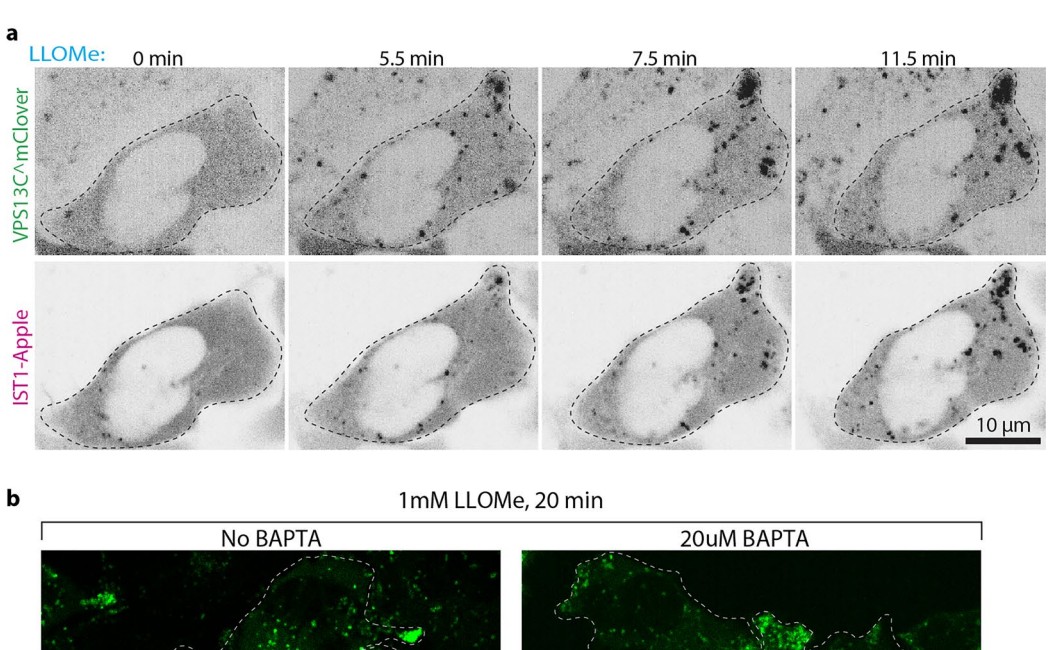

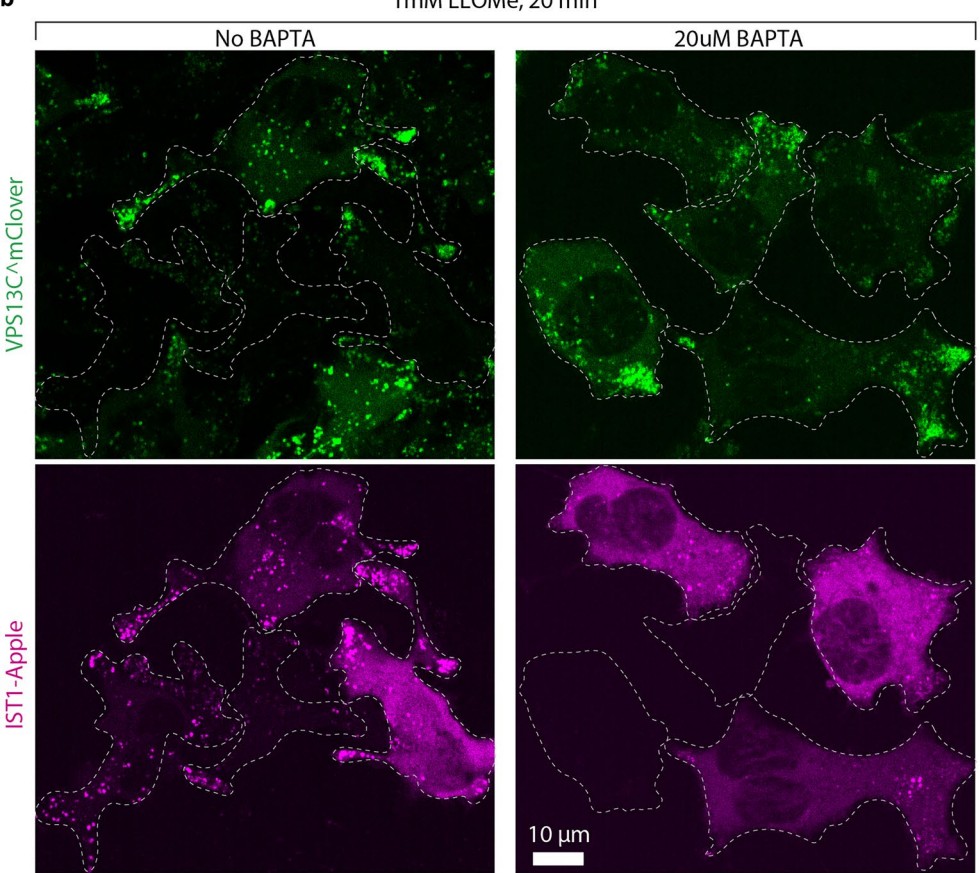

**Extended Data Fig. 2 | LLOMe-induced lysosomal recruitment of VPS13C precedes of ESCRT-III. a**, Time-series of live fluorescence images of VPS13C^mClover-Flp-In cells also co-expressing IST1-Apple, upon 1 mM LLOMe treatment. Note the delayed recruitment of IST1 to lysosomes relative to VPS13C. Individual channel images are shown as inverted greys. The experiment was repeated three times with similar results. **b**, Live fluorescence images of

VPS13C^mClover-Flp-In cells also co-expressing IST1-Apple, 20 minutes after addition of 1 mM LLOMe. Cells were pre-incubated for 1 hr with or without the calcium chelator BAPTA (20 µM). Recruitment of IST1, but not of VPS13C, is inhibited by BAPTA. The experiment was repeated three times with similar results.

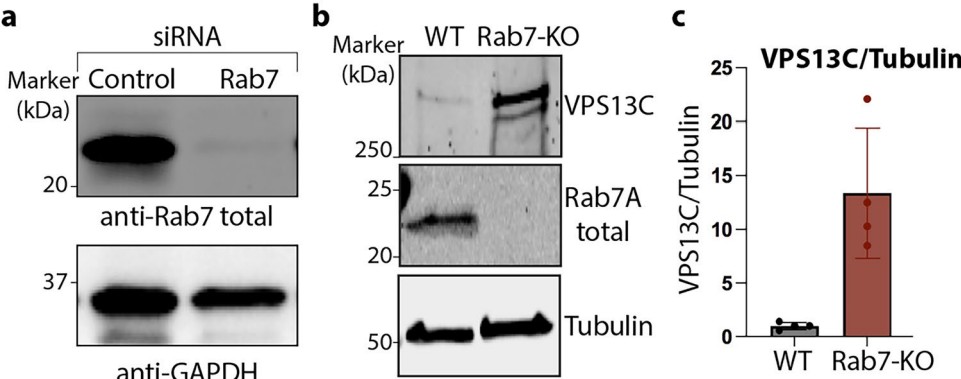

**Extended Data Fig. 3 | Validation of the Rab7 knockdown or knockout cells.**
**a**, Anti-Rab7 western blot of whole cell lysates from control or Rab7 knockdown VPS13C^mClover-Flp-In cells. GAPDH as a loading control. **b**, Anti-Rab7 or anti-VPS13C western blots of whole cell lysates from WT or Rab7 knockout Hela cells.

Tubulin as a loading control. **c**, Quantifications of the blots in **b**. Bars show the normalized value relative to WT. Error bars represent ±SD. n = 4 biological replicates. Numerical data and unprocessed blots are provided.

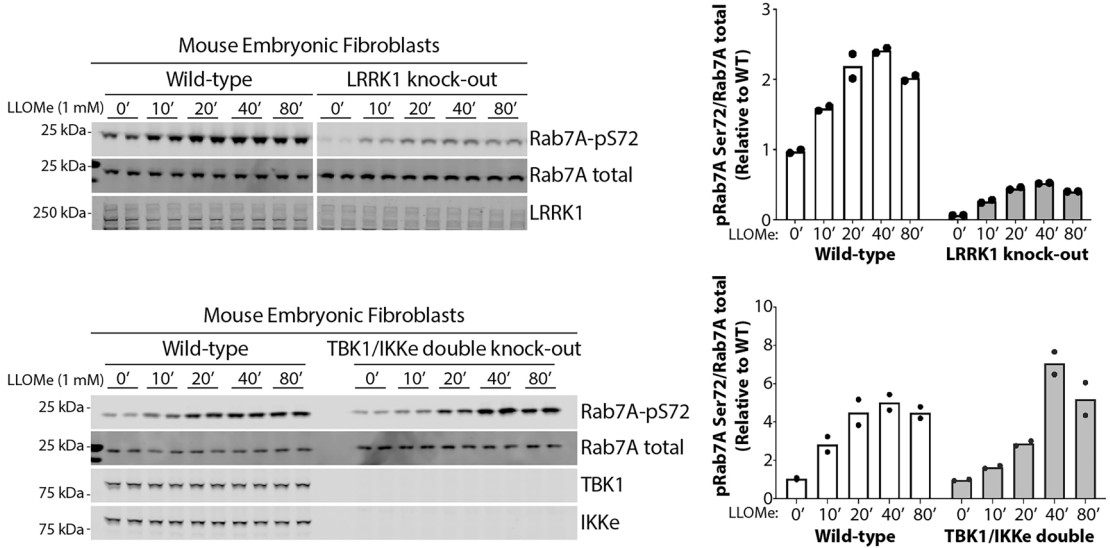

**Extended Data Fig. 4 | Effect of the KO of LRRK1 and TBK1 on LLOMe-induced Rab7 Ser72 phosphorylation in mouse embryonic fibroblasts (MEFs).** Western blot analysis for the indicated proteins in whole cell lysates from the WT, LRRK1 KO or TBK1/IKKe DKO mouse embryonic fibroblasts treated with 1 mM LLOMe for the indicated times. Quantification of the data is shown on the right. n = 2 biological replicates. Numerical data and unprocessed blots are provided.

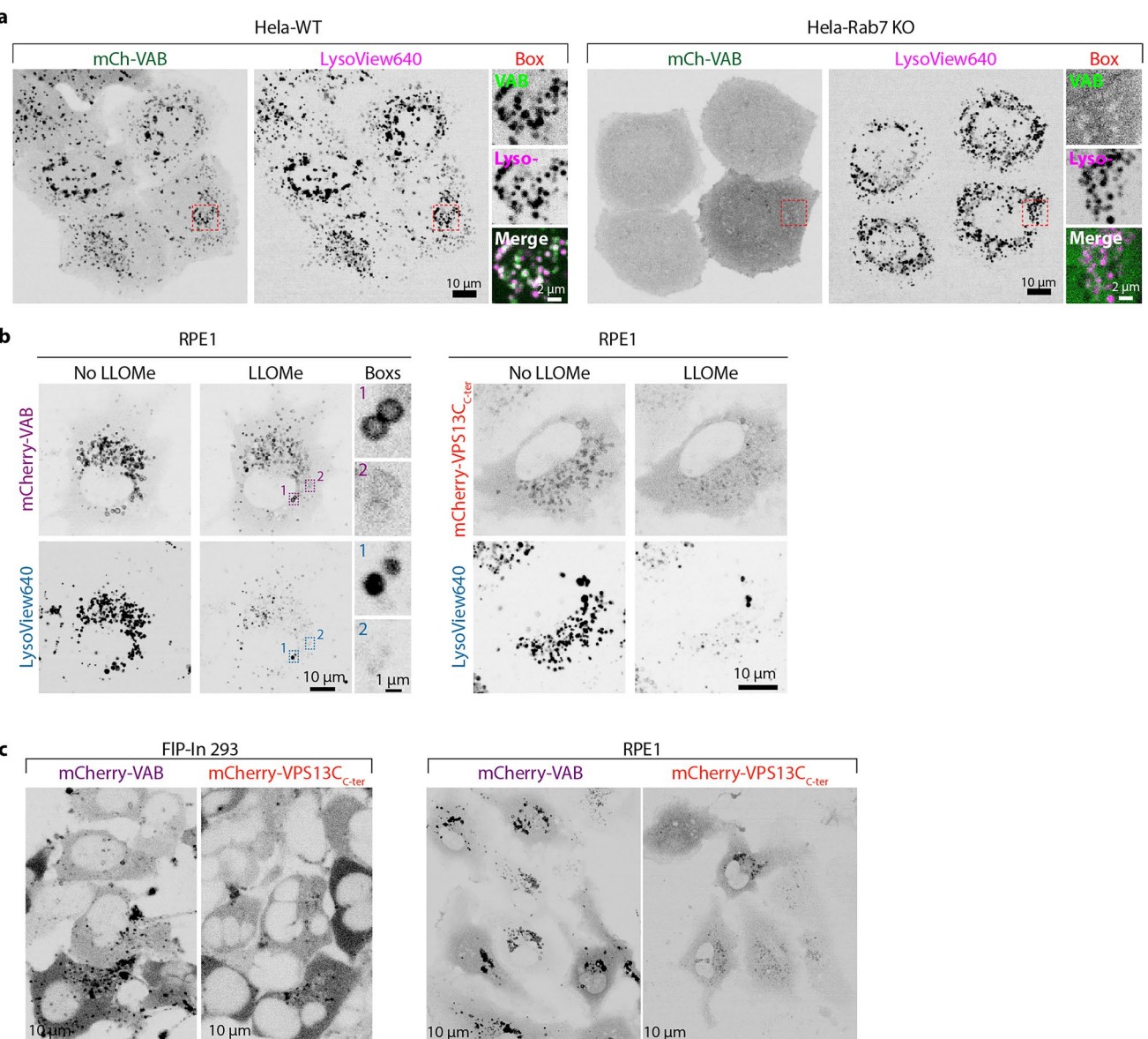

**Extended Data Fig. 5 | Differential lysosome binding of the VAB domain alone and of the entire C-terminal fragment of VPS13C (VPS13C_{C-ter}). a**, Live fluorescence images of WT or Rab7 KO Hela cells expressing mCherry-VAB and labelled with LysoView 640 under basal conditions. Boxed regions are shown on the right. **b**, Live fluorescence images of RPE1 cells expressing exogenous mCherry-VAB (left) or mCherry-VPS13C_{C-ter} (right) before and after 1 mM LLOMe treatment. **c**, Live fluorescence images show localizations of mCherry-VAB or mCherry-VPS13C_{C-ter} in VPS13C^mClover-Flp-In cells (left) or in RPE1 cells (right) in the absence of LLOMe treatment. Note the more prominent lysosome localization of the VAB domain relative to VPS13C_{C-ter}. Individual channel images are shown as inverted greys. Experiments were repeated three times with similar results.

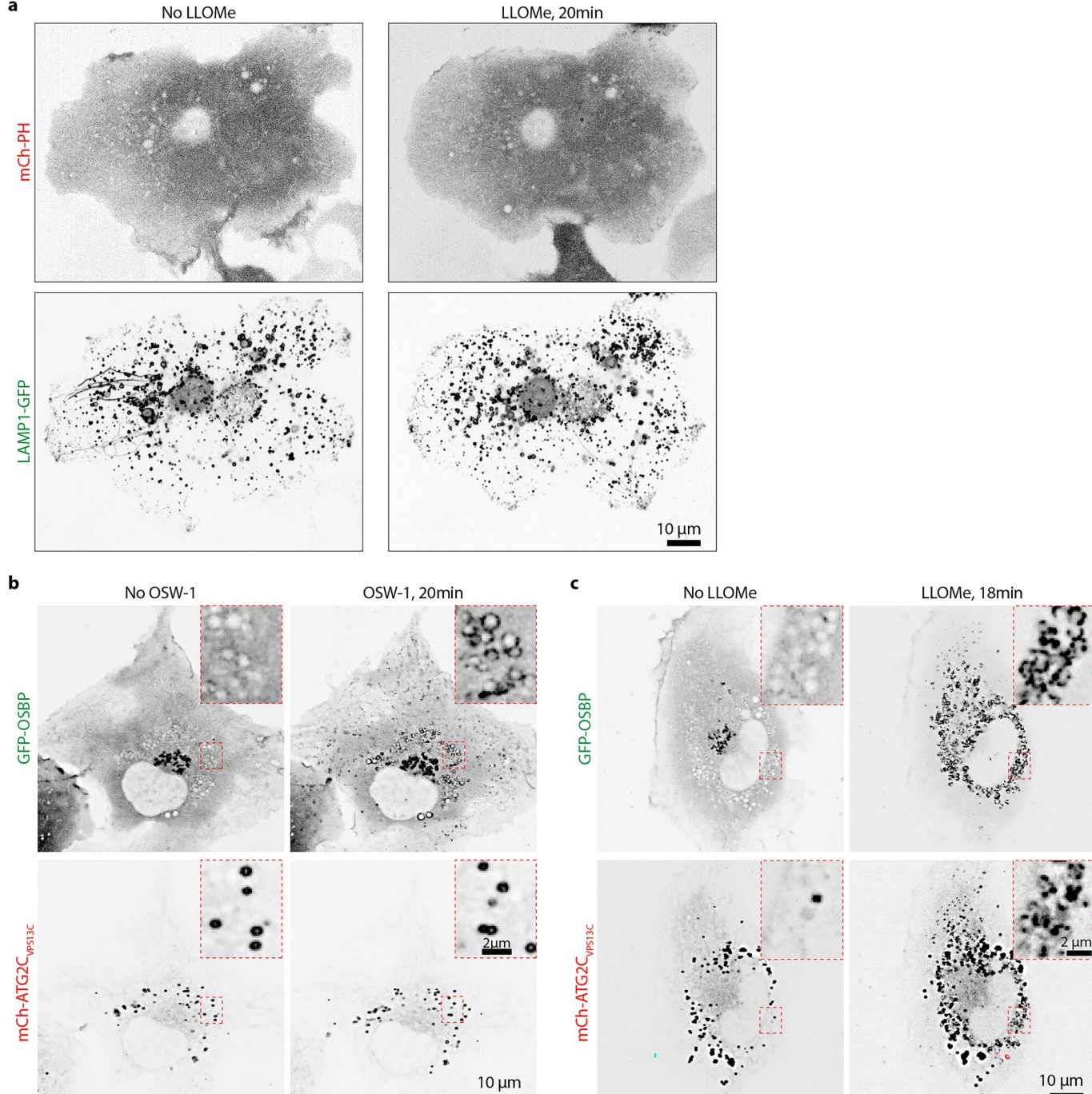

**Extended Data Fig. 6 | Impact of either LLOMe or OSW1 on the localization of the ATG2C domain or the PH domain of VPS13C. a**, Live fluorescence images of RPE1 cells expressing mCherry-PH and LAMP1-GFP before and after 1 mM LLOMe addition. **b**,**c**, Live fluorescence images of RPE1 cells expressing mCherry-ATG2C$_{VPS13C}$ and GFP-OSBP before and after addition of 20 nM OSW1 (**b**) or 1 mM LLOMe (**c**). Black dots visible before LLOMe are lipid droplets that move slightly in position after LLOMe. Individual channel images are shown as inverted greys. Experiments were repeated three times with similar results.

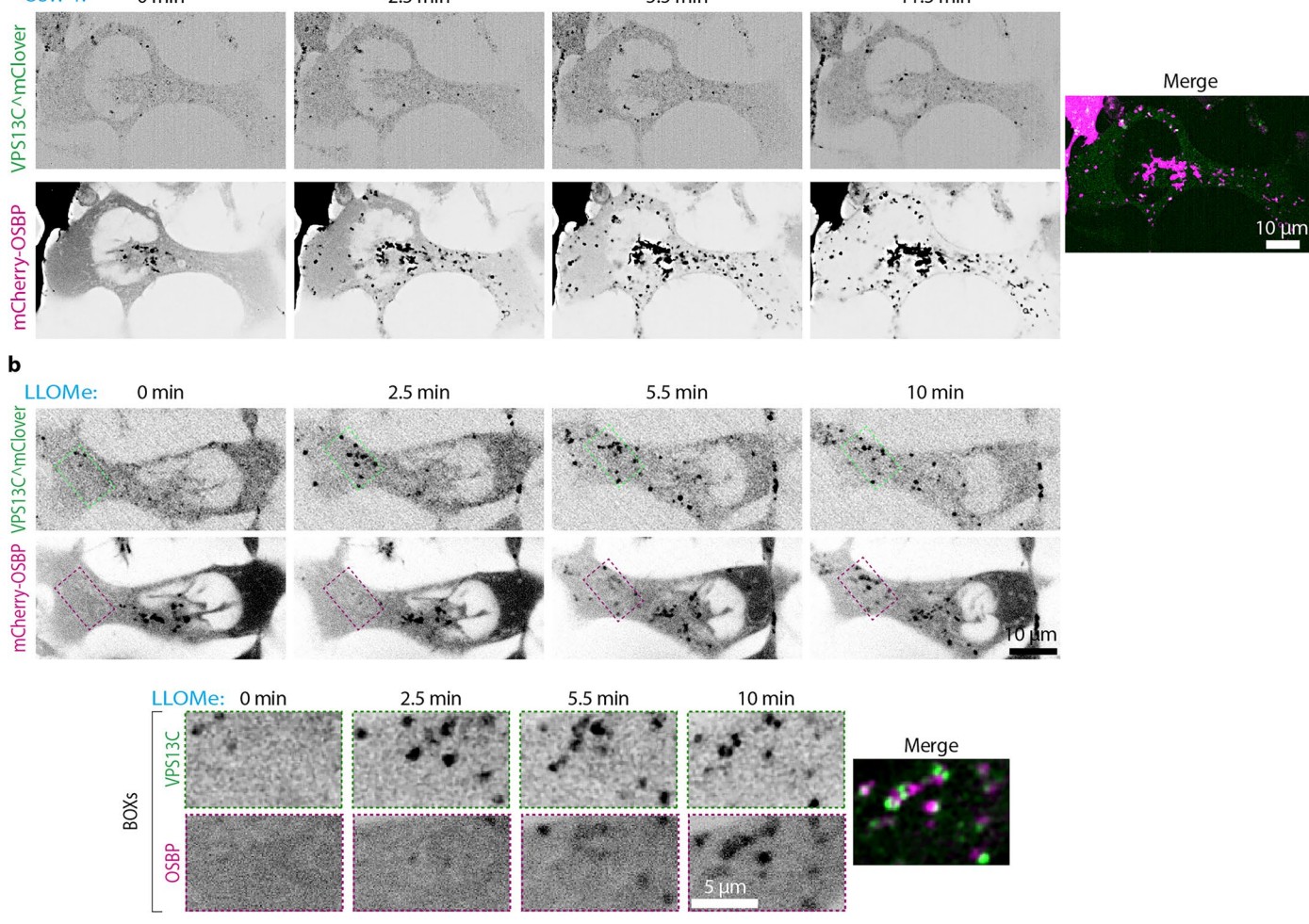

**Extended Data Fig. 7 | Impact of either LLOMe or OSW1 on the localization of VPS13C or OSBP. a**, Time-series of live fluorescence images of VPS13C^mClover-Flp-In cells also co-expressing exogenous mCherry-OSBP, upon 20 nM OSW-1 treatment to induce PI4P accumulation on endolysosomes. Individual channel images are shown as inverted greys. Note that the recruitment of OSBP is not accompanied by the recruitment of VPS13C. **b**, Time-series of live fluorescence images of VPS13C^mClover-Flp-In cells also co-expressing exogenous mCherry-OSBP, upon 1 mM LLOMe treatment. Note the delayed recruitment of OSBP to lysosomes relative to VPS13C. Individual channel images are shown as inverted greys. Boxed regions are shown at the bottom. Individual channel images are shown as inverted greys. Experiments were repeated three times with similar results.

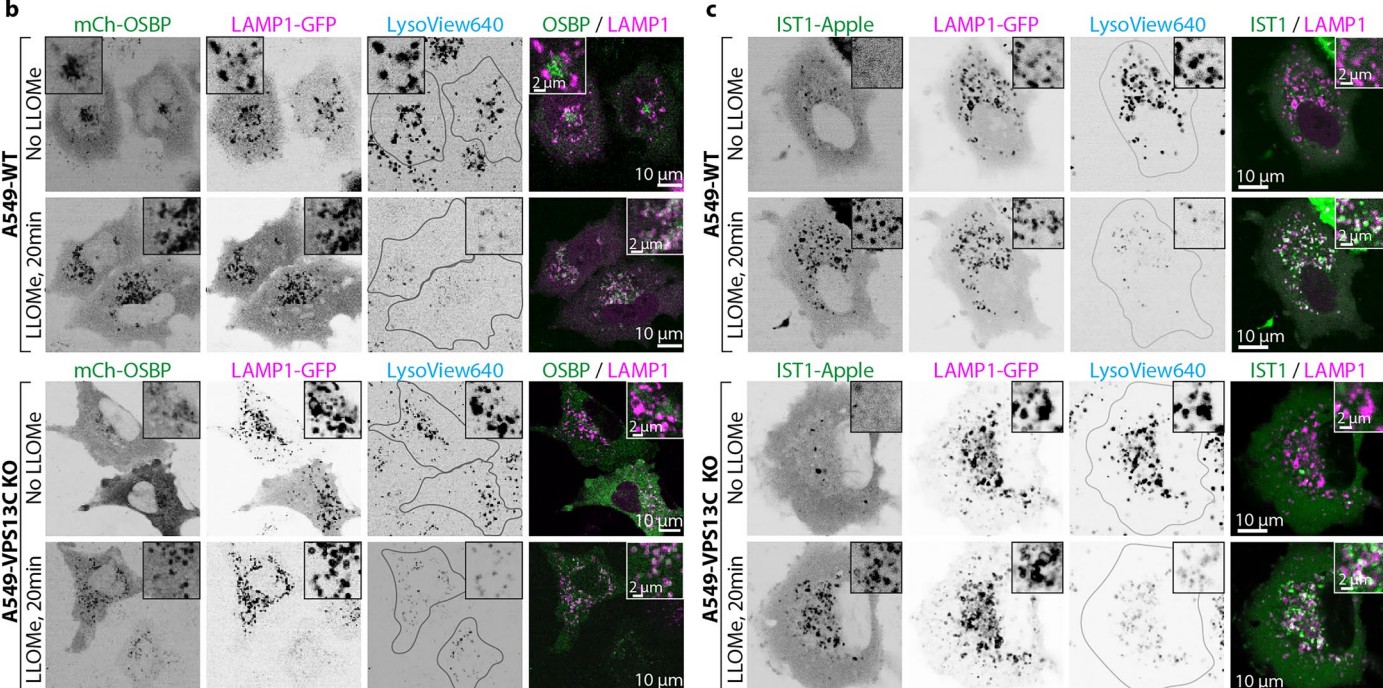

**Extended Data Fig. 8 | Depletion of VPS13C does not affect the lysosomal recruitment of OSBP and IST1. a**, Anti-VPS13C western blot of whole cell lysates from WT or VPS13C knockout A549 cells. Tubulin was used as a loading control. **b,c**, Live fluorescence images of the WT and VPS13C-KO A549 cells expressing mCherry-OSBP and LAMP1-GFP (**b**) or IST1-Apple and LAMP1-GFP (**c**) and stained with LysoView640 before and after 1 mM LLOMe treatment. Individual channel images are shown as inverted greys. Experiments were repeated three times with similar results. Unprocessed blots are provided.

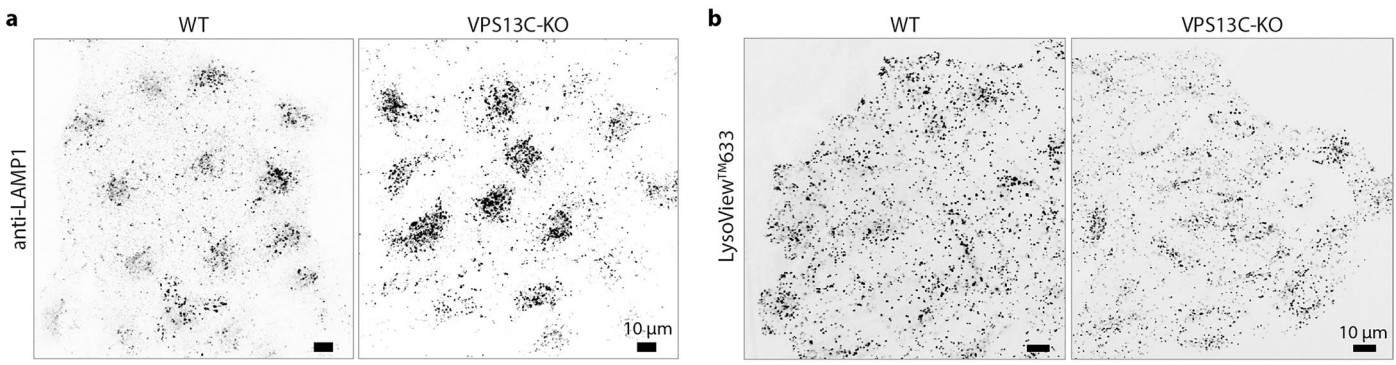

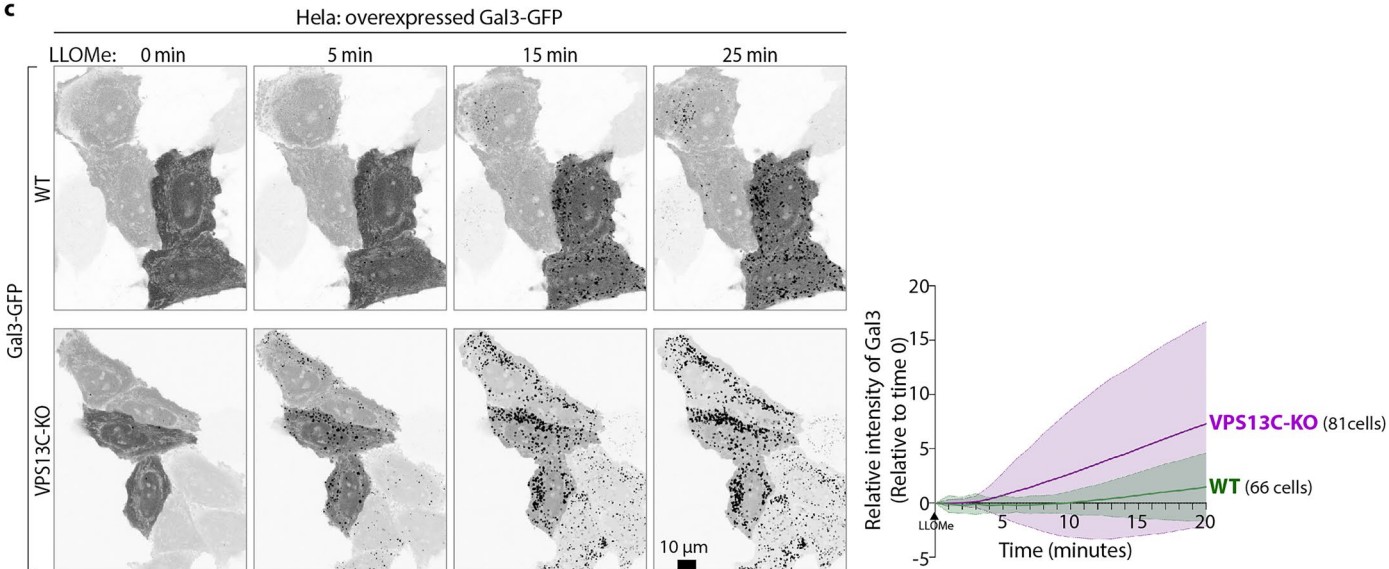

**Extended Data Fig. 9 | Depletion of VPS13C causes disruption of lysosome homeostasis. a**, Anti-LAMP1 immunofluorescence of WT or VPS13C-KO A549 cells. **b**, Live fluorescence images of WT or VPS13C-KO A549 cells incubated with the pH sensitive LysoView633 showing lower fluorescence (higher pH) of the KO cells. **c**, Time-series of live fluorescence images of WT or VPS13C-KO Hela cells expressing Gal3-GFP showing recruitment of Gal3 to damaged lysosomes upon 1 mM LLOMe treatment. Quantification of the intensity of the Gal3-GFP punctate fluorescence per cell after addition of 1 mM LLOMe is shown on the right. n = 66 cells (WT), n = 81 cells (VPS13C-KO) collected from three biological replicates. Shaded areas, mean ± SD. Individual channel images are shown as inverted greys. Experiments were repeated three times with similar results. Numerical data are provided.

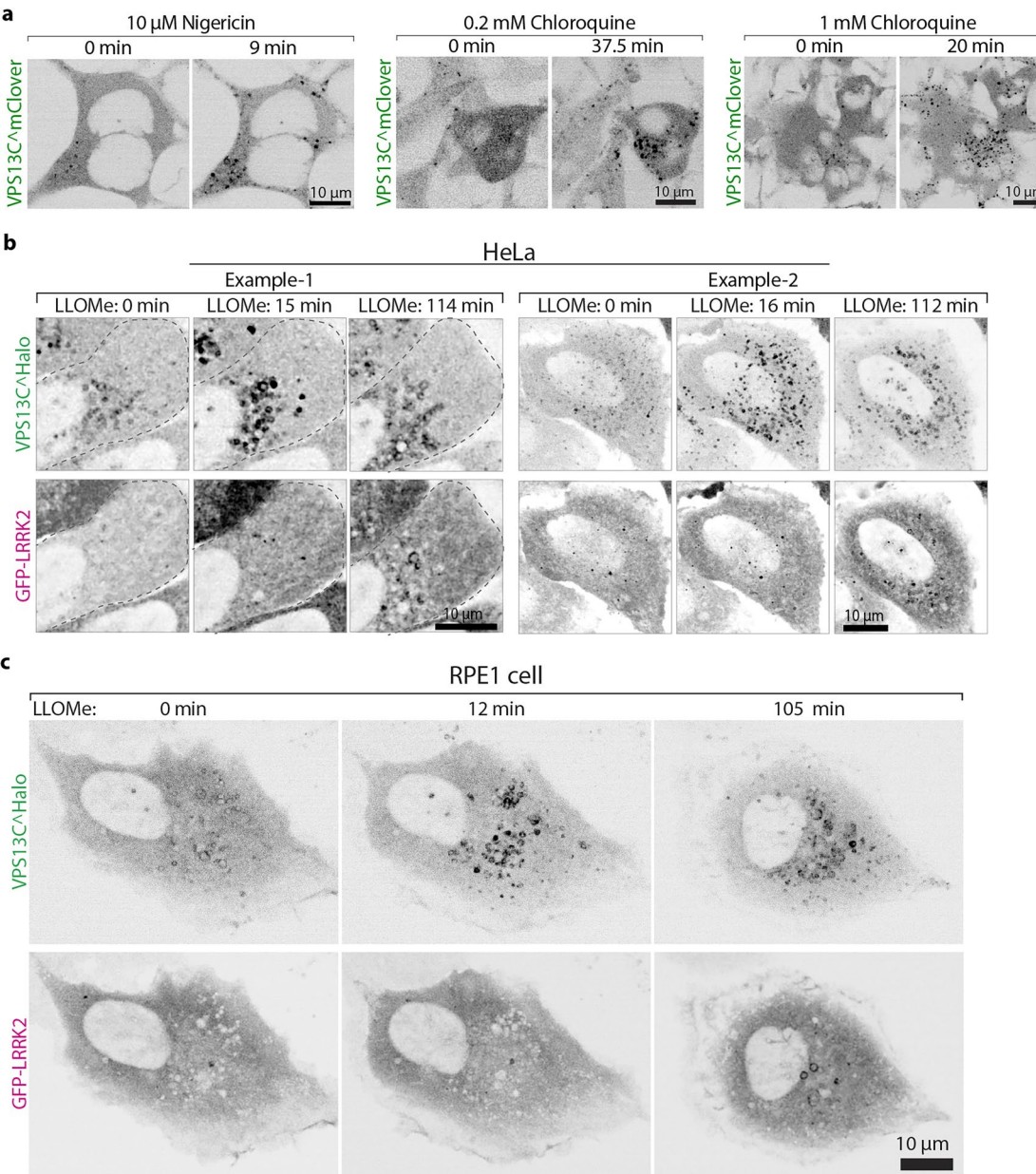

**Extended Data Fig. 10 | Effect of various agents on lysosomal recruitment of VPS13C or LRRK2. a**, Live fluorescence images of VPS13C^mClover-Flp-In cells before and after addition of nigericin or chloroquine, showing lysosomal recruitment of VPS13C. **b**, Time-series of live fluorescence images of Hela cells co-expressing VPS13C^Halo and GFP-LRRK2, showing recruitment of VPS13C and LRRK2 to lysosomes upon 1 mM LLOMe treatment. **c**, Time-series of live fluorescence images of a RPE1 cell co-expressing VPS13C^Halo and GFP-LRRK2 showing recruitment of VPS13C and LRRK2 to damaged lysosomes upon 1 mM LLOMe treatment. LRRK2 is recruited only to a small subset of lysosomes and with a much slower time course than VPS13C. Individual channel images are shown as inverted greys. Experiments were repeated three times with similar results.

# Reporting Summary

## Statistics

For all statistical analyses, confirm that the following items are present in the figure legend, table legend, main text, or Methods section.

| n/a | Confirmed | |
|---|---|---|
| ☐ | ☒ | The exact sample size (*n*) for each experimental group/condition, given as a discrete number and unit of measurement |
| ☐ | ☒ | A statement on whether measurements were taken from distinct samples or whether the same sample was measured repeatedly |
| ☐ | ☒ | The statistical test(s) used AND whether they are one- or two-sided *Only common tests should be described solely by name; describe more complex techniques in the Methods section.* |
| ☒ | ☐ | A description of all covariates tested |
| ☒ | ☐ | A description of any assumptions or corrections, such as tests of normality and adjustment for multiple comparisons |
| ☐ | ☒ | A full description of the statistical parameters including central tendency (e.g. means) or other basic estimates (e.g. regression coefficient) AND variation (e.g. standard deviation) or associated estimates of uncertainty (e.g. confidence intervals) |
| ☐ | ☒ | For null hypothesis testing, the test statistic (e.g. *F*, *t*, *r*) with confidence intervals, effect sizes, degrees of freedom and *P* value noted *Give P values as exact values whenever suitable.* |
| ☒ | ☐ | For Bayesian analysis, information on the choice of priors and Markov chain Monte Carlo settings |
| ☒ | ☐ | For hierarchical and complex designs, identification of the appropriate level for tests and full reporting of outcomes |
| ☒ | ☐ | Estimates of effect sizes (e.g. Cohen's *d*, Pearson's *r*), indicating how they were calculated |

*Our web collection on statistics for biologists contains articles on many of the points above.*

## Software and code

Policy information about availability of computer code

| Data collection | Western blots were imaged on the Odyssey imaging system (LI-COR, ODY-2461); Fluorescence images were captured using a Nikon microscope (Yokogawa CSU-W1 SoRa, Nikon) |
|---|---|
| Data analysis | All immunoblot data and fluorescence images were processed and analyzed with ImageJ/Fiji (Version 2.14.0/1.54f). Statistical analysis was carried out with GraphPad Prism version 8.0.1 |

For manuscripts utilizing custom algorithms or software that are central to the research but not yet described in published literature, software must be made available to editors and reviewers. We strongly encourage code deposition in a community repository (e.g. GitHub). See the Nature Portfolio guidelines for submitting code & software for further information.

## Data

Policy information about availability of data

All manuscripts must include a data availability statement. This statement should provide the following information, where applicable:
- Accession codes, unique identifiers, or web links for publicly available datasets
- A description of any restrictions on data availability
- For clinical datasets or third party data, please ensure that the statement adheres to our policy

Data is available on Zenodo, DOI:10.5281/zenodo.14846056

# Research involving human participants, their data, or biological material

Policy information about studies with [human participants or human data](). See also policy information about [sex, gender (identity/presentation), and sexual orientation]() and [race, ethnicity and racism]().

| | |
|---|---|
| Reporting on sex and gender | n/a |
| Reporting on race, ethnicity, or other socially relevant groupings | n/a |
| Population characteristics | n/a |
| Recruitment | n/a |
| Ethics oversight | n/a |

Note that full information on the approval of the study protocol must also be provided in the manuscript.

# Field-specific reporting

Please select the one below that is the best fit for your research. If you are not sure, read the appropriate sections before making your selection.

☒ Life sciences    ☐ Behavioural & social sciences    ☐ Ecological, evolutionary & environmental sciences

For a reference copy of the document with all sections, see [nature.com/documents/nr-reporting-summary-flat.pdf](http://nature.com/documents/nr-reporting-summary-flat.pdf)

# Life sciences study design

All studies must disclose on these points even when the disclosure is negative.

| | |
|---|---|
| Sample size | No statistical methods were used to predetermine sample sizes but our sample sizes are similar to those reported in previous studies in this field. The sample size and replications for each experiment is specified in the figure legends. |
| Data exclusions | No data were excluded. |
| Replication | All experiments were performed at least in triplicate unless indicated otherwise in the figure legends. |
| Randomization | For fluorescence microscopy image acquisition, cells were selected at random microscopy fields. |
| Blinding | Data analysis was not performed blind to the experimental conditions. Most cell-based quantifications were conducted using live-cell imaging from time-lapse movies, ensuring internal controls. Additionally, the quantification methods relied on software-based automatic calculations. |

# Reporting for specific materials, systems and methods

We require information from authors about some types of materials, experimental systems and methods used in many studies. Here, indicate whether each material, system or method listed is relevant to your study. If you are not sure if a list item applies to your research, read the appropriate section before selecting a response.

## Materials & experimental systems

| n/a | Involved in the study |
|---|---|
| ☐ | ☒ Antibodies |
| ☐ | ☒ Eukaryotic cell lines |
| ☒ | ☐ Palaeontology and archaeology |
| ☐ | ☒ Animals and other organisms |
| ☒ | ☐ Clinical data |
| ☒ | ☐ Dual use research of concern |
| ☒ | ☐ Plants |

## Methods

| n/a | Involved in the study |
|---|---|
| ☒ | ☐ ChIP-seq |
| ☒ | ☐ Flow cytometry |
| ☒ | ☐ MRI-based neuroimaging |

## Antibodies

| | |
|---|---|
| Antibodies used | Anti-LAMP1 (Cell Signaling Technology, 9091; RRID: AB_2687579, for WB, 1:2000)<br>anti-LAMP1 (Abcam, ab25630; RRID:AB_470708, for IF, 1:100)<br>anti-Galectin3 (R&D Systems, IC1154G; RRID:AB_10890949, for IF, 1:50) |

anti-VPS13C (Proteintech, 29844-1-AP; RRID: AB_3086177, for WB, 1:1000)
anti-GM130 (BD Biosciences, 610822; RRID: AB_398141, for WB, 1:2000)
anti-PDI (CST, 2446S; RRID: AB_2298935, for WB, 1:1000)
anti-VAPB (Sigma-Aldrich, HPA013144; RRID: AB_1858717, for WB 1:4000)
anti-Rab7 (Cell Signaling Technology, 9367; RRID: AB_1904103, for WB, 1:1000)
anti-Rab7A (Sigma Aldrich, R8779; RRID:AB_609910, for WB: 1:2000)
anti-pSer72 Rab7(Abcam, ab302494; RRID: AB_2933985, for WB, 1:1000)
anti-GAPDH (Proteus, 40-1246; for WB, 1:1000)
anti-GFP (Abcam, ab290; RRID: AB_303395, for WB, 1:1000)
anti-tubulin (Sigma Aldrich, T5168; RRID: AB_477579, for WB, 1:2000)
anti-mCherry (Abcam, Ab125096; RRID: AB_11133266, for WB, 1:1000)
anti-LRRK1 (MRC Reagents and Services, S405C; for WB: 1 µg/mL)
anti-IKKe (Cell Signaling Technology, 3416S; for WB: 1:2000)
anti-TBK1 (Cell Signaling Technology, 3504S; RRID: AB_2255663, for WB, 1:2000)
anti-pSer172 TBK1 (Cell Signaling Technology, 5483S; RRID: AB_10693472, for WB, 1:1000)

Validation

All antibodies used in this study were commercially obtained and validated either by the suppliers, previous studies, or this study. Validation was based on factors such as the molecular weight of detected bands, the use of samples from knockout or knockdown cells or mice, and the reproducibility of results.

Anti-LAMP1 (Cell Signaling Technology, 9091; RRID: AB_2687579)-PMID:37487100; Vendor:https://www.cellsignal.com/products/primary-antibodies/lamp1-d2d11-xp-rabbit-mab/9091?srsltid=AfmBOor2rf0T6V7V_SF7MJ-VwEkEmKGP5wdIHSi6b1ZSEYrBmrn1O9nr
anti-LAMP1 (Abcam, ab25630; RRID:AB_470708)-PMID: 38020041; Vendor:https://www.abcam.com/en-us/products/primary-antibodies/lamp1-antibody-h4a3-ab25630?srsltid=AfmBOopuGERLMpIH-CAFS6UpjtwGVuCg01tX0sKyRfivIqkowmgua_bb
anti-Galectin3 (R&D Systems, IC1154G; RRID:AB_10890949)-PMID: 30314966; Vendor:https://www.rndsystems.com/products/human-galectin-3-alexa-fluor-488-conjugated-antibody_ic1154g
anti-VPS13C (Proteintech, 29844-1-AP; RRID: AB_3086177)-PMID: 35657605; Vendor:https://www.ptglab.com/products/VPS13C-Antibody-29844-1-AP.htm?srsltid=AfmBOoo7mg91LAtgih8Uf9MdJSgPwYX4F62oWOBTRcHm-EXmgkLlCw59
anti-GM130 (BD Biosciences, 610822; RRID: AB_398141)-PMID: 35650196; Vendor:https://www.bdbiosciences.com/en-us/products/reagents/microscopy-imaging-reagents/immunofluorescence-reagents/purified-mouse-anti-gm130.610822?tab=product_details
anti-PDI (CST, 2446S; RRID: AB_2298935)-PMID: 39080411; Vendor:https://www.cellsignal.com/products/primary-antibodies/pdi-antibody/2446?srsltid=AfmBOorZt6OJ-7cgnWmLQuih3acQCQt14XDH9qdW6SwTxUlVT6wy1QZE
anti-VAPB (Sigma-Aldrich, HPA013144; RRID: AB_1858717)-PMID:37528084; Vendor:https://www.sigmaaldrich.com/US/en/product/sigma/hpa013144?srsltid=AfmBOor3xkJB5wqu0yyUwFUniVVjhTm_x89uOG7qM5W1rwK-iMQAm2ft
anti-Rab7 (Cell Signaling Technology, 9367; RRID: AB_1904103)-PMID: 37141099; Vendor:https://www.cellsignal.com/products/primary-antibodies/rab7-d95f2-xp-rabbit-mab/9367?srsltid=AfmBOooOm-51XTvpHu3vtk-u9Er9N1M5JoWIC3kmCYL9IhfeUCG7qMCv
anti-Rab7A (Sigma Aldrich, R8779; RRID:AB_609910)-PMID: 24145164; Vendor:https://www.sigmaaldrich.com/US/en/product/sigma/r8779?srsltid=AfmBOorcquFbixad2UwJ6cX_Z4fZzGPL4K7hAFT1LJOBV6Wmt1o6ztEM
anti-pSer72 Rab7(Abcam, ab302494; RRID: AB_2933985)-PMID: 37141099; Vendor:https://www.abcam.com/en-us/products/primary-antibodies/rab7-phospho-s72-antibody-mjf-r38-1-ab302494
srsltid=AfmBOop8aqeHGAmnyDRlHDvu_JxVABZ5WHWknoLOl3r7DNBj6U-Px5Fv
anti-GAPDH (Proteus, 40-1246)-PMID: 39331042;
anti-GFP (Abcam, ab290; RRID: AB_303395); Vendor:https://www.abcam.com/en-us/products/primary-antibodies/gfp-antibody-ab290?srsltid=AfmBOoofiEkHQYrBdlGtQYifQBBASyBj6N5Ev8YQZVA2iy4cHxXfze5O
anti-tubulin (Sigma Aldrich, T5168; RRID: AB_477579)-PMID: 39386594; Vendor:https://www.sigmaaldrich.com/US/en/search/t5168?focus=products&page=1&perpage=30&sort=relevance&term=t5168&type=product
anti-mCherry (Abcam, Ab125096; RRID: AB_11133266)-PMID: 27447450; Vendor:https://www.abcam.com/en-us/products/primary-antibodies/mcherry-antibody-1c51-ab125096?srsltid=AfmBOop5kY_4E_vgDxPlMXAifkvtNuo9AW6Kz2BFusNV4_75j7XDPjEv
anti-LRRK1 (MRC Reagents and Services, S405C)-PMID: 33459343; Vendor:https://mrcppureagents.dundee.ac.uk/reagents-view-antibodies/588116
anti-IKKe (Cell Signaling Technology, 3416S)-PMID: 37595039; Vendor:https://www.cellsignal.com/products/primary-antibodies/ikke-d61f9-xp-rabbit-mab/3416?srsltid=AfmBOoqB2t9sqU2w8PD0ufk0O46318GWXTUuXk--5xzFUe796J73TDvx
anti-TBK1 (Cell Signaling Technology, 3504S; RRID: AB_2255663)-PMID: 37595039; Vendor:https://www.cellsignal.com/products/primary-antibodies/tbk1-nak-d1b4-rabbit-mab/3504?srsltid=AfmBOoo2vLldCH6jHEY4x-xr-4h6E_6fKnKlr2Dw9fz5qU55A4pIf3fT
anti-pSer172 TBK1 (Cell Signaling Technology, 5483S; RRID: AB_10693472)-PMID: 39712456; Vendor:https://www.cellsignal.com/products/primary-antibodies/phospho-tbk1-nak-ser172-d52c2-xp-rabbit-mab/5483?srsltid=AfmBOop2-rUaa7Jn0erfxFOZtgWbifhqglJIsb47EBUazZUraG5qSNPE

# Eukaryotic cell lines

Policy information about cell lines and Sex and Gender in Research

Cell line source(s)

RPE1 cells (ATCC, RRID:CVCL_4388); HeLa (RRID:CVCL_R965); Flp-In TREx 293 (Invitrogen); A549 cells (ATCC, RRID:CVCL_0023)

Authentication

Cell lines obtained were not authenticated

Mycoplasma contamination

Cells were tested negative for mycoplasma contamination using PCR-based assays (Lonza Mycoplasma kit) .

Commonly misidentified lines
(See ICLAC register)

No commonly misidentified cell lines was used.

# Animals and other research organisms

Policy information about studies involving animals; ARRIVE guidelines recommended for reporting animal research, and Sex and Gender in Research

| | |
|---|---|
| Laboratory animals | Mice were maintained under specific pathogen-free conditions and housed at an ambient temperature (20–24°C) and humidity (45–55%) and maintained on a 12 h light/12 h dark cycle, with free access to food (SDS RM No. 3 autoclavable) and water. |
| Wild animals | No wild animals were used in this study |
| Reporting on sex | The gender was not considered in this study |
| Field-collected samples | No filed collected samples in this study |
| Ethics oversight | Mice studies were ethically reviewed and carried out in accordance with Animals (Scientific Procedures) Act 1986 and regulations set by the University of Dundee and the U.K. Home Office. All mice studies and breeding were approved by the University of Dundee ethical committee and performed under a U.K. Home Office project license. |

Note that full information on the approval of the study protocol must also be provided in the manuscript.

# Plants

| | |
|---|---|
| Seed stocks | n/a |
| Novel plant genotypes | n/a |
| Authentication | n/a |

