## [Peer Review File · Nature Cell Biology]

The bridge-like lipid transport protein VPS13C/PARK23 mediates ER-lysosome contacts upon lysosome damage

Corresponding Author: Dr Pietro De Camilli

Version 0:

Decision Letter:

Dear Pietro,

Thank you again for submitting your manuscript "Lysosome damage triggers acute formation of ER to lysosomes membrane tethers mediated by the bridge-like lipid transport protein VPS13C", to Nature Cell Biology. Thank you also for your patience with the peer review process. Your manuscript has now been seen by 3 referees, who are experts in lysosome damage/repair (Referee #1); lipid transport (Referee #2); and lysosome damage/repair (Referee #3), and whose comments are pasted below. In light of their advice, we regret that we cannot offer to publish the study in Nature Cell Biology.

As you will see, although the reviewers found the work interesting, they raised serious concerns that question the conceptual advance that these findings represent over previous work and the strength of the data and of the novel conclusions that can be drawn at this stage. In particular, the reviewers felt that the study lacked functional insights into the role of VPS13C in the repair process, and they also indicated that a greater understanding of how VPS13C fits with the other known repair pathways and determinants would be important. We have discussed their concerns, which we find significant - we appreciate that they would require a considerable amount of new experimentation and that this is not trivial. Overall, we have regrettably concluded that the reviewers' concerns are too significant to move forward with the manuscript at the journal.

Although we are very sorry that we cannot offer to publish your manuscript, please let me know if you would like me to consult my colleagues at our sister journals, such as Nature Communications. Alternatively, to transfer your manuscript directly, please use our manuscript transfer portal. You will not have to re-supply manuscript metadata and files, unless you wish to make modifications. For more information, please see our [manuscript transfer FAQ](http://www.nature.com/authors/author_resources/transfer_manuscripts.html?WT.mc_id=EMI_NPG_1511_AUTHORTRANSF&WT.ec_id=AUTHOR) page.

We are very sorry that we could not be more positive on this occasion, but we thank you for the opportunity to consider this work. We also hope you find the reviews below useful as you decide on the next steps for the manuscript.

With kind regards,
Melina

Melina Casadio, PhD
Senior Editor, Nature Cell Biology
ORCID ID: <https://orcid.org/0000-0003-2389-2243>

Reviewers' comments:

Reviewer #1 (Remarks to the Author):

This manuscript describes the recruitment of the lipid channel protein VPS13C to damaged lysosomes. Recruitment requires interaction between the VAB domain of VPS13C and the small GTPase Rab7, proposedly as consequence of release of intramolecular steric hindrance. VPS13C is recruited to damaged lysosomes with faster kinetics than another PD-associated protein, LRRK2. It is suggested that lipid delivery to lysosomes by VPS13C is part of an early response to lysosome damage. These are interesting data, but in the absence of any biological function for VPS13C on damaged lysosomes they remain entirely descriptive. Novelty is limited by the previous discovery that another lipid channel, ATG2A, is recruited to damaged lysosomes for lipid transfer from the ER. The interaction between Rab7 and the VAB domain of VPS13C was also described before.

Detailed comments:

1. Previous work (reference 30) has shown that the lipid channel ATG2A is recruited to damaged lysosomes as part of the PITT pathway. Since ATG2A and VPS13C are known to have overlapping functions in autophagy, it is hardly surprising that they are both recruited to damaged lysosomes.
2. It is disappointing that the function of VPS13C on damaged lysosomes has not been investigated. Is it involved in lysosome repair, resilience to damage, or other aspects of lysosome biology?
3. It is speculated that release of an intramolecular brake exposes the VAB domain so that it can bind to Rab7. What triggers this release (apparently not Rab7 phosphorylation), and which rearrangements in VPS13C occur prior to Rab7 binding?
4. The genetic association of VPS13C with PD is highlighted, but there is nothing in the current manuscript that sheds any light on causal mechanisms of PD.

Reviewer #2 (Remarks to the Author):

This manuscript from Wang et al describes a rapid recruitment of the Vps13C lipid transport protein to lysosomes in response to lysosomal damaging agents. As Vps13C can link the lysosomal membrane to the ER, this recruitment results in association of the damaged lysosomes with ER membranes, perhaps as a source of lipids for damage repair via the Vps13C transporter. Mutations in VPS13C are associated with increased risk of Parkinson's disease and Lrrk2, another Parkinson's-associated protein, also translocates to lysosomes in response to damage. The authors show that the recruitments of Vps13C and Lrrk2 are distinct both in terms of timing and genetic requirements. Similarly, Vps13C recruitment is distinct from that of OSBP, a different lipid transporter previously shown to translocate to damaged lysosomes. Thus, the work identifies a novel response to lysosomal stress, dysfunction of which may be linked to Parkinson's.

Major comments:

The data presented show that Vps13C recruitment occurs quite rapidly and is distinct from previously reported pathways. However, the significance of this recruitment and how (or if) it is linked to lysosomal repair or to the recruitment of other factors, remains unclear. This could be explored with some straightforward experiments in Vps13C knockdown cells. For instance, the authors use Gal3 recruitment to the lysosome as a marker for lysosome permeabilization occurring after persistent damage. Does the timing of Gal3 recruitment speed up in Vps13C knockdown? If so, that would provide evidence that Vps13C recruitment promotes repair.

Further, the authors have shown that lysosomal translocation of Vps13C precedes several other responses. While Vps13C and LRRK2 translocation are clearly independent events, it could be examined whether lysosomal recruitment of ESCRT or OSBP is affected in vps13C knockdown cells.

The quality of the data presented is very high. However, the quantity of cells examined is quite low (Figure 1, 21 cells; Figure 2, 15 cells; Figure 4, 20 cells; Figure 5, ">15 cells") and it's not clear how many times each experiment was performed. For Figure 3, each experiment was apparently performed only once, and no indication is given of how many cells were examined to generate the representative images shown. Both number of cells and number of experiments should be reported in each case, and to be confident in the result, every experiment needs to have been done at least twice.

Reviewer #3 (Remarks to the Author):

A. The manuscript by Wang et al., indicates that VPS13C is recruited to damaged lysosomes and bridges the damaged organelle with the ER for repair. The authors show that this recruitment is Rab7-dependent and that different mechanisms activate the later proteins recruited to lysosomes such as LRRK2. The paper is exceptionally well written, especially the discussion.

B. The area of study is significant. On the weaknesses analysis side, there is some sense of the lessened novelty in a lipid transfer protein recruitment to damaged lysosomes, although the specifics in this case warrant attention and further study.

C,E. The abstract underscores this and an impression of a limited forward progress, using statements "most likely" (for intramolecular inhibition) and "putative role" (for VPS13C in lipid transfer). Furthermore, Rab7 action, LRRK1 and 2 timing, are touched upon but with mostly negative findings, leaving the reader with the sense that much more experimental work needs to be carried out and that the mechanisms are yet to be defined. Nevertheless, this reviewer wishes to emphasize the overall impression being a positive one, as this reviewer views the negative findings of authors' explorations of the relationships as just as valuable as if there was a clear demonstration of mechanisms, although it would have been preferred to have at least one of those resolved with some certainty.

Specific points:

1. In most of the figures with VPS13C^{ΔmClover}, the protein is accumulating in peripheral lysosomes of the cells, no matter how long has been the LLOMe treatment. This suggests that some subpopulations of endolysosomal organelles are affected

and not lysosomes in general, as presented. Do all lysosomal populations have the capacity to recruit VPS13C, or it is just the peripheral endosomes? Is the above phenomenon related to the proposed autoinhibition state within VPS13C in different areas of the cell, and how does that spatial selectivity, if indeed not an artifact, works?

2. In the first subheading, the authors mention that the levels of endogenous plus exogenous VPS13C are equal in WT and the stable VPS13C cells, once decreased the endogenous (Figure S1B). How the authors calculate that? Besides, why there is a reduction on the endogenous expression? While this is considered to be a “plus” by the authors, this could be taken as an indication of the plasmid affecting the cell in other ways, which needs to be understood if the point is to be made. Why did the authors choose 24h instead of 48h? Does this level of expression change overtime?

3. In Figure 1, the authors addressed the recruitment of VPS13C to damaged lysosomes after LLOMe treatment by microscopy and western blot of the purified endolysosomes. In Figure 1E, the authors showed that VPS13C is increased in LLOMe-treated cells as the VAPB also increased, suggesting the recruitment of the ER protein to the endolysosomal organelles. However other common markers are not shown. Have they checked other markers as Rab7, Rab5 or even other lysosomal damage markers like Galectin 3 or Ubiquitin?

4. Furthermore what is the role of the lipid transfer proteins ATG2, previously reported in Nature to be key for lysosomal repair?

5. Is there an interplay between VPS13C and ATG2s in this?

6. In Figure S1C, the authors pointed out that the use of E64d impaired recruitment of VPS13C in LLOMe-treated cells. Why did authors choose to present only intensity of the VPS13C puncta and is not showing also puncta quantification. The levels and intensity of Control non treated (top left image) VPS13C^{mClover} seem bigger than 0 relative intensity.

7. In Figures 2C-E, the authors pointed out that Rab7 expression interfered in VPS13C recruitment, but the levels of the protein seem already low without any treatment. Have they checked the levels of VPS13C by western blot in siRNA Rab7? Once in Figure S1B the authors already showed that endogenous expression of the protein is reduced when overexpress with mClover, is it possible that the knockdown is also affecting the whole expression too?

8. In Figure S3, the authors also checked phosphorylation of LRRK2 and Rab10. Why did the authors choose those proteins and MLi2 to test? LRRK1? In the subheading “The access of full length VPS13C to Rab7 on endolysosomes undergoes regulation”, the authors checked whether the VPS13C is recruited by activated Rab7. The rationale for this is unclear. How can Rab7 binding to VAB domain regulate VPS13C recruitment in LLOMe-treated cells? The activation state seems not to matter, but the indirect analysis of those by the mCherry-VAPB still doesn't show Rab7 in this context. Have the authors checked by other assays the association?

9. In figure S5, VPS13C seems to not be involved in the PITT pathway. Interestingly, another lipid transfer protein, ATG2, is also involved in PITT pathway and its recruitment is dependent on PS and cholesterol. Did the authors check if in the absence of ATG2, VPS13C could act as the other lipid transfer to repair the lysosome in a similar model in PITT pathway?

10. In Figure 5, the authors checked if CASM inducers activate the early recruitment of VPS13C to the lysosomes, and the Salip and SopF (alone) were not able to recruit VPS13C. Why did the assembly/disassembly and the regulation of the V-ATPase activity affect VPS13C?

11. In the well-written discussion, the authors suggest that VPS13C could provide lipid transport in the early stages connecting ER to lysosomes, like what is happening in PITT pathway. Is there any aspect of lipid transfer that the authors can demonstrate in an assay to reinforce this point?

Minor points:

1. The title is awkward: Lysosome damage triggers acute formation of ER to lysosomes membrane tethers mediated by the bridge-like lipid transport protein VPS13C

2. If the authors show merged images in Figures: 1G, 2B, S5A, S5B could help the readers understand the colocalization of the markers used.

3. In Figures 1D, 2D, the authors didn't mention the time of the treatment with LLOMe.

4. In Figure 4C, the authors mislabeled the figures – panel is written mCh-VAB and the legend mScarlet PPM1H.

5. In page 8, first paragraph, there is “(see above)” making it difficult for the reader to understand the context besides going back and forth, and is ambiguous and left to the reader's choice to find the pertinent argument or not.

6. Lysosomal repair references could be more inclusive.

**For Nature Portfolio general information and news for authors, see <http://npg.nature.com/authors>.

Version 1:

Decision Letter:

Dear Pietro,

Thank you for your email asking us to reconsider our decision on your manuscript, “The bridge-like lipid transport protein VPS13C/PARK23 mediates acute formation of ER to lysosome tethers in response to lysosome damage”. We are always willing to hear the authors' perspective, but we must first prioritize decisions on new submissions. We appreciate your patience while we considered this appeal.

I have now discussed your manuscript, the referees' comments, and your rebuttal, in detail with my colleagues, and we

would be willing to re-review the revised manuscript provided the following files are also provided for peer review, and that nothing similar is accepted for publication at Nature Cell Biology or published elsewhere in the meantime.

In particular, please provide, in addition to the revised manuscript and point-by-point response to the full reviews, verbatim (we request the following files for all revised manuscripts):

- the completed Editorial Policy Checklist (found here <https://www.nature.com/documents/nr-editorial-policy-checklist.pdf>), and Reporting Summary (found here <https://www.nature.com/documents/nr-reporting-summary.pdf>). This is essential for reconsideration of the manuscript and these documents will be available to editors and referees in the event of peer review. For more information see below. Please also ensure that the presentation of statistical information in the revised submission complies with Nature Cell Biology's statistical guidelines (see below).

Please use the link below to submit the complete manuscript files and include a point-by-point response to the complete reviewer comments, verbatim as provided in their reports.

Link Redacted

Please let us know how you wish to proceed and when we can expect your revised manuscript.

With kind regards,

Melina

Melina Casadio, PhD
Senior Editor, Nature Cell Biology
Consulting Editor, Nature Structural & Molecular Biology
ORCID ID: <https://orcid.org/0000-0003-2389-2243>

GUIDELINES FOR EXPERIMENTAL AND STATISTICAL REPORTING

REPORTING REQUIREMENTS – To improve the quality of methods and statistics reporting in our papers we have recently revised the reporting checklist we introduced in 2013. We are now asking all life sciences authors to complete two items: an Editorial Policy Checklist (found here <https://www.nature.com/documents/nr-editorial-policy-checklist.pdf>) that verifies compliance with all required editorial policies and a reporting summary (found here <https://www.nature.com/documents/nr-reporting-summary.pdf>) that collects information on experimental design and reagents. These documents are available to referees to aid the evaluation of the manuscript. Please note that these forms are dynamic 'smart pdfs' and must therefore be downloaded and completed in Adobe Reader. We will then flatten them for ease of use by the reviewers. If you would like to reference the guidance text as you complete the template, please access these flattened versions at <http://www.nature.com/authors/policies/availability.html>.

We strongly recommend the presentation of source data for graphical and statistical analyses as a separate Supplementary Table, and request that source data for all independent repeats are provided when representative experiments of multiple independent repeats, or averages of two independent experiments are presented. This supplementary table should be in Excel format, with data for different figures provided as different sheets within a single Excel file. It should be labelled and

numbered as one of the supplementary tables, titled "Statistics Source Data", and mentioned in all relevant figure legends.

Version 2:

Decision Letter:

Our ref: NCB-A54381B

22nd January 2025

Dear Pietro,

Thank you for submitting your revised manuscript "The bridge-like lipid transport protein VPS13C/PARK23 mediates acute formation of ER to lysosome tethers in response to lysosome damage" (NCB-A54381B). It has now been seen by the original referees and their comments are below. The reviewers find that the paper has been strengthened in revision, and therefore we'll be happy in principle to publish it in Nature Cell Biology, pending minor revisions to satisfy the referees' final requests and to comply with our editorial and formatting guidelines. We appreciated your responses to their points and have decided that it will be reasonable not to include the ATG2 data in your final manuscript. Please still provide textual responses to all reviewer comments with your final resubmission once it's ready and please address the other reviewer points with edits to the manuscript text.

Of note, the current version of your manuscript is in a PDF format. We apologize as we cannot proceed with PDFs at this stage. Could you please email us a copy of the file in an editable format (Microsoft Word or LaTeX)? Thank you in advance.

Once we receive the Word file, we will begin performing detailed checks on your paper and will send you a checklist detailing our editorial and formatting requirements in about 2 weeks. Please do not upload the final materials and make any revisions until you receive this additional information from us.

Lastly, I should share I am temporarily moving to Nature Structural & Molecular Biology (NSMB) full-time this month. I have been a consulting editor there since the fall and I just started as the locum chief editor while their Chief editor is on parental leave. *I will be back at Nature Cell Biology full-time in the fall of 2025.*

I am very sorry as this means I won't handle your manuscript further myself; however, it will be in brilliant hands with the rest of the NCB team, who knows the study from our team-wide discussions of it and has access to the full history and all my notes. I look forward to seeing it progress through the next stages from afar and thank you again for your efforts in revision & for considering NCB for the work.

Thank you again for your interest in Nature Cell Biology. Please do not hesitate to contact me if you have any questions.

Sincerely,

Melina

Melina Casadio, PhD
Senior Editor, Nature Cell Biology
Consulting Editor, Nature Structural & Molecular Biology
ORCID ID: <https://orcid.org/0000-0003-2389-2243>

Reviewer #1 (Remarks to the Author):

The authors have successfully addressed most of the points I raised, but in the rebuttal they mention that they do not observe any recruitment of ATG2A to damaged lysosomes. This is an important piece of information that needs to be described explicitly in the manuscript, and the authors should include Fig. 1 for reviewers as a supplementary figure in their manuscript.

Reviewer #2 (Remarks to the Author):

In the revision, the authors have thoroughly addressed my concerns with the first version of the manuscript. In addition to a convincing description of the behavior of Vps13C in response to lysosomal damage, they now supply evidence of the biological significance of this relocalization. In addition, they have further clarified the relationship between Vps13 and other lysosomal damage responses in the cell as well as provided further evidence for auto-inhibitory regulation of Vps13C. This latter result has potential implications for other Vps13 family proteins.

Reviewer #3 (Remarks to the Author):

The authors have taken several of the reviewers' comments into consideration and carefully revised their study. The

improvements are noticeable and this reviewer considers the amendments potentially appropriate and acknowledges a mechanistic progress in Rab7-VPS13C recruitment.

Before fully endorsing this study, this reviewer feels that the following remains and should be addressed:

1. The insistence on the absence of ATG2 is concerning as this has been seen not only in the original Nature publication by Tan but also in other follow-up studies that remain to be cited.
2. Along the same lines, if this claim is to be made a comprehensive panel of relevant positive controls where ATG2 can be observed by the authors needs to be included to demonstrate that absence is not due to technical difficulties.
3. The referencing still appears to be very selective. For example, while the authors use Gal3 they ignore the literature on functional and other aspects of Gal3 in lysosome damage repair (it's not just a marker). This needs to be remedied.
4. In some of the answers to the specific questions by this reviewer, the authors have brushed aside the recommendation saying that this is beyond the scope. Perhaps it is not so, or else the question would have not been posed. A different treatment/explanation of the topic (which could be mentioned in discussion) would have been a preferred amendment.

ADDITIONAL COMMENT FROM REV#3 ABOUT POINT #2 ABOVE

My comment was to provide a positive comparator control (e.g. detect ATG2 in another known role for example on autophagosomes) to their claim that they do not detect the previously published (by others) ATG2 on lysosomes upon damage.

Version 3:

Decision Letter:

Dear Pietro,

I am pleased to inform you that your manuscript, "The bridge-like lipid transport protein VPS13C/PARK23 mediates ER-lysosome contacts upon lysosome damage", has now been accepted for publication in Nature Cell Biology.

You may wish to make your media relations office aware of your accepted publication, in case they consider it appropriate to organize some internal or external publicity. Once your paper has been scheduled you will receive an email confirming the publication details. This is normally 3-4 working days in advance of publication. If you need additional notice of the date and time of publication, please let the production team know when you receive the proof of your article to ensure there is sufficient time to coordinate. Further information on our embargo policies can be found here:

<https://www.nature.com/authors/policies/embargo.html>

Please note that *Nature Cell Biology* is a Transformative Journal (TJ). Authors may publish their research with us through the traditional subscription access route or make their paper immediately open access through payment of an article-processing charge (APC). Authors will not be required to make a final decision about access to their article until it has been accepted. [Find out more about Transformative Journals](https://www.springernature.com/gp/open-research/transformative-journals)

If you have not already done so, we strongly recommend that you upload the step-by-step protocols used in this manuscript to protocols.io (<https://protocols.io>), an open online resource that allows researchers to share their detailed experimental know-how. All uploaded protocols are made freely available and are assigned DOIs for ease of citation. Protocols and Nature Portfolio journal papers in which they are used can be linked to one another, and this link is clearly and prominently visible in the online versions of both. Authors who performed the specific experiments can act as primary authors for the Protocol as they will be best placed to share the methodology details, but the Corresponding Author of the present research paper should be included as one of the authors. By uploading your Protocols onto protocols.io, you are enabling researchers to more readily reproduce or adapt the methodology you use, as well as increasing the visibility of your protocols and papers. You can also establish a dedicated workspace to collect your lab Protocols. Further information can be found at <https://www.protocols.io/help/publish-articles>.

Nature Cell Biology encourages authors presenting evidence for cell, biological, molecular, and genetic interactions to consider communicating these findings using Biofactoid (<https://biofactoid.org/>). This tool helps users share a searchable representation of interactions (e.g. binding, gene expression, post-translational modification) between genes, gene products, or chemicals. Information added to Biofactoid, with author attribution, is shared on social media and public databases, such as Pathway Commons, where it can be discovered and analyzed in the context of a large and growing corpus of knowledge.

Best regards,

George

George Inglis, PhD
Senior Editor

[Research Cross-Journal Editorial Team](https://www.nature.com/ncb/research-cross-journal-editorial-team)
Nature Cell Biology

** Visit the Springer Nature Editorial and Publishing website at [www.springernature.com/editorial-and-publishing-jobs](http://editorial-jobs.springernature.com?utm_source=ejp_NCB_email&utm_medium=ejp_NCB_email&utm_campaign=ejp_NCB) for more information about our career opportunities. If you have any questions please click [here](mailto:editorial.publishing.jobs@springernature.com). **

Reviewers' comments:

Reviewer #1 (Remarks to the Author):

This manuscript describes the recruitment of the lipid channel protein VPS13C to damaged lysosomes. Recruitment requires interaction between the VAB domain of VPS13C and the small GTPase Rab7, proposedly as consequence of release of intramolecular steric hindrance. VPS13C is recruited to damaged lysosomes with faster kinetics than another PD-associated protein, LRRK2. It is suggested that lipid delivery to lysosomes by VPS13C is part of an early response to lysosome damage. These are interesting data, but in the absence of any biological function for VPS13C on damaged lysosomes they remain entirely descriptive. Novelty is limited by the previous discovery that another lipid channel, ATG2A, is recruited to damaged lysosomes for lipid transfer from the ER. The interaction between Rab7 and the VAB domain of VPS13C was also described before.

We thank the reviewer for raising these points. Concerning absence of evidence of any biological function of VPS13C, we have now provided direct evidence for increased lysosome fragility of VPS13C KO cells using the Galectin 3 assay (Fig. 6 for A549 VPS13C KO cells and Fig. S9C for VPS13C KO HeLa cells). Additionally, we found that lysosomal acidification was compromised in VPS13C KO cells (Fig. 6). These new results corroborate what we had reported previously: that absence of VPS13C in HeLa cells results in activation of the cGAS-STING innate immunity pathway and that such activation may be explained by leaky and functionally defective lysosomes (PMID: 35657605).

This reviewer has also stated that the reported acute recruitment of ATG2A (another bridge-like lipid transport protein) to damaged lysosomes limits the novelty of our study. The reviewer most likely refers to the papers 1) by Tan and Finkel (PMID: 36071159) which is based on microscopic analysis and on studies of Galectin3-based assays of lysosome fragility of ATG2 KO cells, and 2) by Cross et al (PMID: 37796195), which demonstrated biochemically (but not microscopically) formation of an LC3-ATG2 interaction in response to LLOMe. In our lab, however, we have not observed recruitment of ATG2A to lysosomes in response to LLOMe using several cell types. A representative example of the difference between ATG2 and VPS13C recruitment is shown below in Fig. 1. We were surprised by this result as the region of VPS13C which we have found to be responsible for its recruitment to lysosomes in response to LLOMe share structural homology to a corresponding region in ATG2 (hence its name ATG2C domain), yet we have found that the two proteins (VPS13C and ATG2), and even their isolated ATG2C domains, behave differently. We cannot explain the discrepancy and we prefer not to address it in this manuscript. Regardless of the questions concerning ATG2 (which are outside the focus of our paper), we have generated important new data for VPS13C in the context of lysosome damage that stands on its own.

Figure 1: ATG2A is not recruited to LLOMe-induced damaged lysosomes. Live fluorescence images of Hek293 cells expressing mCherry-ATG2a and VPS13C^{mClover} before or after 1 mM LLOMe treatment. No lysosome recruitment of mCherry-ATG2a, but robust recruitment of VPS13C^{mClover} was observed.

Concerning the interaction between Rab7 and the VAB domain of VPS13C. Yes, this interaction was described before. The point of the present paper is not to describe the interaction. The important and central point of our paper is to show how such interaction is regulated and that binding of full length VPS13C to Rab7 in living cells requires the lysosome damage-induced release of an autoinhibition.

Detailed comments:

1. Previous work (reference 30) has shown that the lipid channel ATG2A is recruited to damaged lysosomes as part of the PITT pathway. Since ATG2A and VPS13C are known to have overlapping functions in autophagy, it is hardly surprising that they are both recruited to damaged lysosomes.

We have addressed above the comment on ATG2A.

2. It is disappointing that the function of VPS13C on damaged lysosomes has not been investigated. Is it involved in lysosome repair, resilience to damage, or other aspects of lysosome biology?

We had already published a study on the impact of VPS13C on lysosome function where we had reported several alterations (PMID: 35657605). In such study we had shown a constitutive activation of the cGAS-STING pathway of innate immunity and provided evidence for a role of defective lysosomes in such activation: presence of mitochondrial DNA in the cytosol (consistent with leaky lysosomes containing mitochondrial fragments to be degraded) and defective degradation of STING in lysosomes.

Importantly, in response to this comment of the reviewer, we conducted functional experiments in VPS13C KO cells (Fig. 6), directly demonstrating that the loss of VPS13C increases lysosome fragility, supporting its role in lysosome resilience to damage and consistent with a role in their repair.

3. It is speculated that release of an intramolecular brake exposes the VAB domain so that it can bind to Rab7. What triggers this release (apparently not Rab7 phosphorylation), and which rearrangements in VPS13C occur prior to Rab7 binding?

We have taken to heart this important comment and performed extensive experimentation to address it. Our new results support a model according to which the ATG2C domain of VPS13C acts as a regulatory switch preventing the interaction of the VAB domain of VPS13C with Rab7. Specifically, our new experiments suggest that under basal condition the C-terminal fragment of VPS13C, the ATG2C-PH domain, blocks access of the VAB domain to Rab7. Upon membrane damage/perturbation, ATG2C_{VPS13C} domain binds to damaged membranes resulting in, or stabilizing, a different conformational state of the C-terminal portion of VPS13C. This change exposes the Rab7 binding site on the VAB domain, enabling interaction with Rab7 and thus stabilizing binding of VPS13C to lysosomal membranes. The ATG2C domain of VPS13C comprises a tetrahelix bundle with amphipathic properties, previously shown to account for its robust binding to lipid droplets (PMID: 30093493) whose outer monolayer is thought to expose packing defects (PMID: 29316443). Our results, summarized in new Fig. 5G, are consistent with reports that lipid packing defect induced by lysosome perturbing agents on their membrane are the signals that recruit amphipathic helices (PMID: 38503285, PMID: 38594929).

4. The genetic association of VPS13C with PD is highlighted, but there is nothing in the current manuscript that sheds any light on causal mechanisms of PD.

We have now directly shown that mutations in VPS13C result in increased fragility of lysosomes (faster lysis and accessibility to Gal3 in response to LLOMe), thus providing support to the hypothesis that a lability of lysosomes may be implicated in PD pathogenesis. Our results are convergent with evidence that LRRK2, the most frequently mutated protein in familial PD, may also be implicated in lysosome membrane repair (PMID: 33177079). Fragile lysosomes may leak into the cytosol substances that are toxic for the cell, including mitochondrial RNA and DNA which accumulate in lysosomes as a result of mitophagy, and whose release into the cytosol activate innate immunity (see our previous study of VPS13C KO cells and activation of innate immunity. PMID: 35657605). These considerations obviously do not address the issue of why VPS13C loss-of-function mutations leads to PD and not to a broader

neurodegenerative phenotype. However, the question of how mutations of house-keeping proteins affect selectively brain cell subpopulations is a major open question in the field of neurodegenerative diseases, one that goes much beyond the scope of this manuscript.

Reviewer #2 (Remarks to the Author):

This manuscript from Wang et al describes a rapid recruitment of the Vps13C lipid transport protein to lysosomes in response to lysosomal damaging agents. As Vps13C can link the lysosomal membrane to the ER, this recruitment results in association of the damaged lysosomes with ER membranes, perhaps as a source of lipids for damage repair via the Vps13C transporter. Mutations in VPS13C are associated with increased risk of Parkinson's disease and Lrrk2, another Parkinson's-associated protein, also translocates to lysosomes in response to damage. The authors show that the recruitments of Vps13C and Lrrk2 are distinct both in terms of timing and genetic requirements. Similarly, Vps13C recruitment is distinct from that of OSBP, a different lipid transporter previously shown to translocate to damaged lysosomes. Thus, the work identifies a novel response to lysosomal stress, dysfunction of which may be linked to Parkinson's.

Major comments:

The data presented show that Vps13C recruitment occurs quite rapidly and is distinct from previously reported pathways. However, the significance of this recruitment and how (or if) it is linked to lysosomal repair or to the recruitment of other factors, remains unclear. This could be explored with some straightforward experiments in Vps13C knockdown cells. For instance, the authors use Gal3 recruitment to the lysosome as a marker for lysosome permeabilization occurring after persistent damage. Does the timing of Gal3 recruitment speed up in Vps13C knockdown? If so, that would provide evidence that Vps13C recruitment promotes repair.

We thank the reviewer for raising this important question and for the suggestion. We have now performed these experiments and observed a faster recruitment of galectin 3 to lysosomes of VPS13C KO cells in response LLOMe in two VPS13C KO cell types tested (new Fig. 6C and 6D, and Fig. S9C). We have also shown that lysosomal acidification is compromised in VPS13C KO cells (new Fig. 6B). We consider these findings an important addition to our study and we thank this reviewer for the suggestion.

Further, the authors have shown that lysosomal translocation of Vps13C precedes several other responses. While Vps13C and LRRK2 translocation are clearly independent events, it could be examined whether lysosomal recruitment of ESCRT or OSBP is affected in vps13C knockdown cells.

In response to this comment, we examined the lysosomal recruitment of ESCRT (IST1) and OSBP in both WT and VPS13C KO cells. We did not observe any significant difference, as IST1 and OSBP were still efficiently recruited following LLOMe treatment in the KO cells. These results are shown in new Fig. S8.

The quality of the data presented is very high. However, the quantity of cells examined is quite low (Figure 1, 21 cells; Figure 2, 15 cells; Figure 4, 20 cells; Figure 5, ">15 cells") and it's not clear how many times each experiment was performed. For Figure 3, each experiment was apparently performed only once, and no indication is given of how many cells were examined to generate the representative images shown. Both number of cells and number of experiments should be reported in each case, and to be confident in the result, every experiment needs to have been done at least twice.

For figure 1, 2 and 4, we have increased the number of cells used for the quantification experiments and added the corresponding cell counts in each figure. For Figure 3, we repeated the Western blots multiple times and included the quantified data in the figure. For Figure 5 (now Figure 7) we have performed the experiment additional times and we show two new representative sequences in Fig. S10. Additionally, while we previously stated in the Methods/Live cell imaging section that "All experiments were repeated

at least 3 times, unless otherwise indicated in the figure legends”, in the revised manuscript we have now specified the exact number of replicates in each figure legend.

Reviewer #3 (Remarks to the Author):

A. The manuscript by Wang et al., indicates that VPS13C is recruited to damaged lysosomes and bridges the damaged organelle with the ER for repair. The authors show that this recruitment is Rab7-dependent and that different mechanisms activate the later proteins recruited to lysosomes such as LRRK2. The paper is exceptionally well written, especially the discussion.

We appreciate the reviewer’s positive feedback on the writing of our manuscript.

B. The area of study is significant. On the weakness’s analysis side, there is some sense of the lessened novelty in a lipid transfer protein recruitment to damaged lysosomes, although the specifics in this case warrant attention and further study.

We wish to emphasize that mechanisms of VPS13C recruitment (investigated in greater depth in this revised manuscript than in the original manuscript) are very different from those underlying the recruitment of other lipid transport proteins and that the well-established role of VPS13C in PD adds relevance to our findings. Also concerning novelty: see above our response to the first issue raised by reviewer #1.

C,E. The abstract underscores this and an impression of a limited forward progress, using statements “most likely” (for intramolecular inhibition) and “putative role” (for VPS13C in lipid transfer). Furthermore, Rab7 action, LRRK1 and 2 timing, are touched upon but with mostly negative findings, leaving the reader with the sense that much more experimental work needs to be carried out and that the mechanisms are yet to be defines. Nevertheless, this reviewer wishes to emphasize the overall impression being a positive one, as this reviewer views the negative findings of authors’ explorations of the relationships as just as valuable as if there was a clear demonstration of mechanisms, although it would have been preferred to have at least one of those resolved with some certainty.

First, we would like to thank this reviewer for his/her very positive comments.

We also appreciate his/her comment concerning some of the cautious words that we had chosen to describe our findings. About our statement that an intramolecular inhibition was only a “most likely” possibility to explain regulation of VPS13C recruitment to lysosomes, we addressed this comment frontally with new experiments. Using several new VPS13C deletion constructs and isolated domains, we have now provided compelling evidence for a model in which the C-terminal region of VPS13C occludes the Rab7 binding sites of VPS13C until the ATG2C domain of VPS13C (ATG2C_{VPS13C}) has the opportunity to interact with a damaged lysosome membrane (Fig. 5) (see response to question 3 from Reviewer 1 for details). Thus, we have done precisely what the reviewer had suggested when he/she stated that we should address at least one of the open questions with certainty. Based on these experiments we have removed the words “most likely” from the Abstract and we have replaced the sentence: “This recruitment depends on Rab7 and requires release of a brake, most likely an intramolecular interaction within VPS13C, which hinders access of its VAB domain to lysosome-bound Rab7” with the slightly modified sentence “This recruitment depends on Rab7 and requires a signal at the damaged lysosome surface that releases an inhibited state of VPS13C which hinders access of its VAB domain to lysosome-bound Rab7.” This model is illustrated in new Fig. 5G. About defining VPS13C as a protein with a “putative role” in lipid transfer, we realize that at this point evidence for a role in lipid transfer of the VPS13 protein family is very strong and we removed the word “putative”.

Specific points:

1. In most of the figures with VPS13C^{mClover}, the protein is accumulating in peripheral lysosomes of the cells, no matter how long has been the LLOMe treatment. This suggests that some subpopulations of

endolysosomal organelles are affected and not lysosomes in general, as presented. Do all lysosomal populations have the capacity to recruit VPS13C, or it is just the peripheral endosomes? Is the above phenomenon related to the proposed autoinhibition state within VPS13C in different areas of the cell, and how does that spatial selectivity, if indeed not an artifact, works?

We have not observed a spatial selectivity of VPS13C recruitment to lysosome subpopulations. As we stated in the text relative to Fig. 1E: “Notably, Lamp1-positive lysosomes were concentrated at the tips of cell processes in this cell line, and their localization was not affected by exposure to LLOMe.” In other cell lines such as HeLa (Fig. 2D) and RPE1 (Fig. 4D and E) cells, no peripheral accumulation of VPS13C was observed, as these cells do not exhibit the peripheral concentration of lysosomes.

2. In the first subheading, the authors mention that the levels of endogenous plus exogenous VPS13C are equal in WT and the stable VPS13C cells, once decreased the endogenous (Figure S1B). How the authors calculate that? Besides, why there is a reduction on the endogenous expression? While this is considered to be a “plus” by the authors, this could be taken as an indication of the plasmid affecting the cell in other ways, which needs to be understood if the point is to be made. Why did the authors choose 24h instead of 48h? Does this level of expression change overtime?

Our point here was to simply rule out artifacts due to strong overexpression of VPS13C. So did not carry out a precise quantification of its level as we thought it was not critical to show that overall levels of VPS13C were the same in control and VPS13C^{mClover-Flp-In} cells. The result is highly reproducible as shown below (Fig.2).

Figure 2: Western blots for the indicated proteins of whole cell lysates from VPS13C^{mClover-Flp-In} cells treated with or without tetracycline for 24 hours. Two biological replicates are shown. (The slower migrating band is VPS13C^{mClover} due to its higher MW)

In any case, we have now shortened/simplified the statement relative to Fig. S1B, which now reads: “Anti-VPS13C western blotting of homogenates of these cells revealed that, under the conditions used for our experiments (24 or 48 hours in the presence of 0.1 µg/ml tetracycline), no global overexpression of VPS13C occurred as the levels of endogenous VPS13C were reduced in parallel with expression of exogenous VPS13C^{mClover} (Fig. S1B)”.

For all experiments, except for the Rab7 siRNA knockdown experiments (which require 48 hours to achieve sufficient knockdown), we chose the 24h time point. We did not notice any obvious changes in VPS13C expression between 24 and 48 hours.

3. In Figure 1, the authors addressed the recruitment of VPS13C to damaged lysosomes after LLOMe treatment by microscopy and western blot of the purified endolysosomes. In Figure 1E, the authors showed that VPS13C is increased in LLOMe-treated cells as the VAPB also increased, suggesting the recruitment of the ER protein to the endolysosomal organelles. However other common markers are not shown. Have they checked other markers as Rab7, Rab5 or even other lysosomal damage markers like Galectin 3 or Ubiquitin?

Our primary aim for this experiment was to confirm that also endogenous, untagged VPS13C was recruited to lysosomes, and not only exogenous tagged VPS13C^{mClover}. Lamp1 was used as a control.

An analysis of which other proteins were recruited to LLOMe-damaged lysosomes was beyond the purpose of these studies.

4. Furthermore what is the role of the lipid transfer proteins ATG2, previously reported in Nature to be key for lysosomal repair?

5. Is there an interplay between VPS13C and ATG2s in this?

In response to questions 4 and 5: we are well aware of the study by Tan and Finkel published in Nature (PMID: 36071159) and we have confirmed the very robust PI4P-dependent recruitment of OSBP to lysosomes that they have shown. However, we have attempted to replicate the recruitment of ATG2A to lysosomes in response to LLOMe, but we have not observed such a recruitment in all of the several cell lines that we tested. See also our response to the first comment of reviewer #1. The striking difference between the response of VPS13C and ATG2A is shown above (Fig. 1) for the reviewers.

6. In Figure S1C, the authors pointed out that the use of E64d impaired recruitment of VPS13C in LLOMe-treated cells. Why did authors choose to present only intensity of the VPS13C puncta and is not showing also puncta quantification? The levels and intensity of Control non treated (top left image) VPS13C^{mClover} seem bigger than 0 relative intensity.

We thank the reviewer for bringing this to our attention. In the figure, we presented the data as ‘relative intensity’, as we have normalized the signal in LLOMe-treated cells relative to controls (no LLOMe) at time zero. We have now written “relative to time 0” in the Y axis to enhance understanding. We did not display puncta numbers because, following LLOMe treatment, the intense labeling of lysosomes at sites where they are clustered (at cell tips in the Hek293 cells used for this analysis) makes it nearly impossible to accurately count individual puncta.

7. In Figures 2C-E (Fig. 2C and Fig. S3A in the revised version), the authors pointed out that Rab7 expression interfered in VPS13C recruitment (we guess the reviewer meant that loss of Rab7 impaired VPS13C recruitment), but the levels of the protein seem already low without any treatment. Have they checked the levels of VPS13C by western blot in siRNA Rab7? Once in Figure S1B the authors already showed that endogenous expression of the protein is reduced when overexpress with mClover, is it possible that the knockdown is also affecting the whole expression too?

In response to this question, we have now compared the expression levels of VPS13C in both WT and Rab7 KO cells. As shown in Fig. S3B, the total level of VPS13C is significantly increased, rather than reduced, in the KO cells. This is likely due to compensatory mechanisms aimed at addressing Rab7 KO-induced lysosomal dysfunction.

8. In Figure S3, the authors also checked phosphorylation of LRRK2 and Rab10. Why did the authors choose those proteins and MLI2 to test? LRRK1?

Figure S3 included more information than needed. We have simplified the figure to display only the essential blots and we have deleted the blots which we had included as controls but whose data are peripheral to this study.

In the subheading “The access of full length VPS13C to Rab7 on endolysosomes undergoes regulation”, the authors checked whether the VPS13C is recruited by activated Rab7. The rationale for this is unclear. How can Rab7 binding to VAB domain regulate VPS13C recruitment in LLOMe-treated cells? The activation state seems not to matter, but the indirect analysis of those by the mCherry-VAB still doesn’t show Rab7 in this context. Have the authors checked by other assays the association?

We initially hypothesized that an increase in the activation state of Rab7 could be responsible for the LLOMe-induced recruitment of full-length VPS13C via its VAB domain. However, our findings ruled out this possibility. Instead, our data (including new findings shown in Fig. 5) demonstrate that this interaction depends on the release of an autoinhibitory mechanism within the C-terminal fragment of

VPS13C which exposes the Rab7 binding site on the VAB domain (see response to question 3 from Reviewer 1 for details).

Concerning the interaction between Rab7 and the VAB domain, this was demonstrated in previous studies (PMID: 30093493 and PMID: 35657605). Most importantly, we have now used Rab7 KO cells and show that both the VAB domain and full length VPS13C are no longer recruited to lysosomes in these cells (Fig. S5A).

9. In figure S5, VPS13C seems to not be involved in the PITT pathway. Interestingly, another lipid transfer protein, ATG2, is also involved in PITT pathway and its recruitment is dependent on PS and cholesterol. Did the authors check if in the absence of ATG2, VPS13C could act as the other lipid transfer to repair the lysosome in a similar model in PITT pathway?

We did not investigate VPS13C recruitment in the absence of ATG2. As both proteins are bridge-like lipid transport proteins, they could have overlapping roles, despite being recruited through different mechanisms. However, as we mentioned above at point #5 of this reviewer and in response to the first comment of reviewer #1 (see also Fig. 1), we did not observe an obvious ATG2 recruitment.

10. In Figure 5, the authors checked if CASM inducers activate the early recruitment of VPS13C to the lysosomes, and the Salip and SopF (alone) were not able to recruit VPS13C. Why did the assembly/disassembly and the regulation of the V-ATPase activity affect VPS13C?

We checked a potential role of CASM in VPS13C recruitment as CASM was proposed to mediate the recruitment of other proteins in response to LLOMe. We found this not to be the case. Concerning the question “Why did the assembly/disassembly and the regulation of the V-ATPase activity affect VPS13C?”, we do not understand this question. We did not show that “assembly/disassembly and the regulation of the V-ATPase activity affects VPS13C”.

11. In the well-written discussion, the authors suggest that VPS13C could provide lipid transport in the early stages connecting ER to lysosomes, like what is happening in PITT pathway. Is there any aspect of lipid transfer that the authors can demonstrate in an assay to reinforce this point?

We wish we had a good assay, but we do not, yet. Developing such assays is current major priority in the field.

Minor points:

1. The title is awkward: Lysosome damage triggers acute formation of ER to lysosomes membrane tethers mediated by the bridge-like lipid transport protein VPS13C.

We thank the reviewer for this comment about the title. We have changed it to: “The bridge-like lipid transport protein VPS13C/PARK23 mediates acute formation of ER to lysosome tethers in response to lysosome damage”.

2. If the authors show merged images in Figures: 1G, 2B, S5A, S5B could help the readers understand the colocalization of the markers used.

We have added the merged images as the reviewer suggested.

3. In Figures 1D, 2D, the authors didn't mention the time of the treatment with LLOMe.

We have added the time as suggested.

4. In Figure 4C, the authors mislabeled the figures – panel is written mCh-VAB and the legend mScarleth PPM1H.

We do not understand this comment. There is no mislabeling in this figure. We have double-checked both the figure and the legend to ensure consistency.

5. In page 8, first paragraph, there is “(see above)” making it difficult for the reader to understand the context besides going back and forth, and is ambiguous and left to the reader’s choice to find the pertinent argument or not.

We thank the reviewer for raising this concern. We have modified the text. We deleted “see above” and we have now mentioned Rab7. Old sentence: “These results support the hypothesis (see above) that recruitment of the ER to lysosomes reduces the association with their surface of cytosolic factors by hindering their access to such surface”. New sentence: “These results support the hypothesis that recruitment of the ER to lysosomes reduces the association with their surface of cytosolic factors bound to Rab7 by hindering their access to such surface”.

6. Lysosomal repair references could be more inclusive.

We have added several more references.

February 11, 2025

Rebuttal to the final point #3 and #4 of reviewer #3

#3. The referencing still appears to be very selective. For example, while the authors use Gal3 they ignore the literature on functional and other aspects of Gal3 in lysosome damage repair (it's not just a marker). This needs to be remedied.

We have made the following modification:

Original text: *Expression of fluorescently-tagged galectin3 has been used as a tool to detect severely damaged lysosomes(27, 38). Galectin 3 is a β -galactoside binding lectin that remains soluble in the cytosol of healthy cells. However, it rapidly translocates to damaged lysosomes, where it binds the glycoprotein-rich luminal surface of their membranes (38).*

New text: *“Expression of fluorescently-tagged galectin3 has been used as a tool to detect severely damaged lysosomes(27, 38). Galectin 3 is a β -galactoside binding lectin that remains soluble in the cytosol of healthy cells. However, it rapidly translocates to damaged lysosomes, where it binds the glycoprotein-rich luminal surface of their membranes to help coordinate lysosomal repair and removal processes (38-40).” (the two new references are PMID: 31813797, PMID: 32521192)*

#4. In some of the answers to the specific questions by this reviewer, the authors have brushed aside the recommendation saying that this is beyond the scope. Perhaps it is not so, or else the question would have not been posed. A different treatment/explanation of the topic (which could be mentioned in discussion) would have been a preferred amendment.

The review likely refers to these two sentences of his/her previous review:

Sentence 1:

In Figure 1, the authors addressed the recruitment of VPS13C to damaged lysosomes after LLOMe treatment by microscopy and western blot of the purified endolysosomes. In Figure 1E, the authors showed that VPS13C is increased in LLOMe-treated cells as the VAPB also increased, suggesting the recruitment of the ER protein to the endolysosomal organelles. However other common markers are not shown. Have they checked other markers as Rab7, Rab5 or even other lysosomal damage markers like Galectin 3 or Ubiquitin?

Our manuscript is focused on VPS13C, not globally on mechanisms in lysosome repair, a complex problem that require a multiplicity of factors as many recent studies suggest. In any case made the following change in the discussion

Original text: *While VPS13C recruitment requires Rab7, which binds the VAB domain of VPS13C, it also requires an additional signal. We have ruled out Rab7 phosphorylation, PI4P and Ca^{2+} as the signal.*

New text: *While VPS13C recruitment requires Rab7, which binds the VAB domain of VPS13C, enhanced Rab7 activity at the lysosomes does not appear to be responsible for such recruitment,*

as recruitment of the VAB domain-only construct is not increased by LLOMe. We have also ruled out Rab7 phosphorylation, PI4P and Ca²⁺ as the signal the triggers VPS13C recruitment.

We did show with microscopy that Galectin 3 is recruited and we did not test Rab5, as VPS13C is not recruited to a Rab5 compartment. Concerning ubiquitin, VPS13C does not have ubiquitin binding domain.

Sentence 2. In the well-written discussion, the authors suggest that VPS13C could provide lipid transport in the early stages connecting ER to lysosomes, like what is happening in PITT pathway. Is there any aspect of lipid transfer that the authors can demonstrate in an assay to reinforce this point?

As we stated in the original rebuttal, developing such assays is current major priority in the field. While assays to monitor lipid transport by shuttle-like lipid transport proteins are available and we have extensively used them (PMID: 29222176, PMID: 27065097), no such assay has been developed yet to monitor bridge-like lipid transport in vitro. While we had used fragments of VPS13 to monitor lipid transfer between liposomes (PMID: 30093493), most likely such transport had not occurred by a bridge-like mechanism.